**The EMBO Journal** (2013) 32, 2722–2734
www.embojournal.org

# Endophilin, Lamellipodin, and Mena cooperate to regulate F-actin-dependent EGF-receptor endocytosis

**Anne Vehlow[1], Daniel Soong[1,2], Gema Vizcay-Barrena[3], Cristian Bodo[1], Ah-Lai Law[1], Upamali Perera[1] and Matthias Krause[1,***

[1]King's College London, Randall Division of Cell and Molecular Biophysics, London, UK, [2]King's College London, Cardiovascular Division, British Heart Foundation Centre of Excellence, James Black Centre, London, UK and [3]King's College London, Centre for Ultrastructural Imaging, London, UK

**The epidermal growth factor receptor (EGFR) plays an essential role during development and diseases including cancer. Lamellipodin (Lpd) is known to control lamellipodia protrusion by regulating actin filament elongation via Ena/VASP proteins. However, it is unknown whether this mechanism supports endocytosis of the EGFR. Here, we have identified a novel role for Lpd and Mena in clathrin-mediated endocytosis (CME) of the EGFR. We have discovered that endogenous Lpd is in a complex with the EGFR and Lpd and Mena knockdown impairs EGFR endocytosis. Conversely, overexpressing Lpd substantially increases the EGFR uptake in an F-actin-dependent manner, suggesting that F-actin polymerization is limiting for EGFR uptake. Furthermore, we found that Lpd directly interacts with endophilin, a BAR domain containing protein implicated in vesicle fission. We identified a role for endophilin in EGFR endocytosis, which is mediated by Lpd. Consistently, Lpd localizes to clathrin-coated pits (CCPs) just before vesicle scission and regulates vesicle scission. Our findings suggest a novel mechanism in which Lpd mediates EGFR endocytosis via Mena downstream of endophilin.**

*The EMBO Journal* (2013) **32,** 2722–2734. doi:10.1038/emboj.2013.212; Published online 27 September 2013
*Subject Categories:* membranes & transport
*Keywords:* endophilin; EGF-receptor endocytosis; Ena/VASP proteins; Lamellipodin; Mena

## Introduction

Lpd and RIAM, the two mammalian proteins of the MIG10-RIAM-Lpd (MRL) protein family, harbour several Ena/VASP-binding sites (Krause *et al*, 2004; Lafuente *et al*, 2004). Ena/VASP proteins directly interact with actin to promote the formation of longer, less branched filaments by antagonizing capping activity (Krause *et al*, 2003; Pula and Krause, 2008).

*Corresponding author. Kings College London, Randall Division of Cell and Molecular Biophysics, New Hunt's House, Guy's Campus, London SE1 1UL, UK. Tel.: +44 (0)20 7848 6959; Fax: +44 (0)20 7848 6435; E-mail: matthias.krause@kcl.ac.uk

In contrast, N-WASP activates the Arp2/3 complex to nucleate branched actin filaments (Chesarone and Goode, 2009). Lpd recruits Ena/VASP proteins to the leading edge of cells thereby regulating lamellipodia protrusion, dorsal ruffling of fibroblasts, axon extension, and branching of neurons (Krause *et al*, 2004; Michael *et al*, 2010).

Although F-actin polymerization is required for endocytosis in yeast, a role for the actin cytoskeleton during clathrin-mediated endocytosis (CME) in mammalian cells is controversial (Lamaze *et al*, 1997; Fujimoto *et al*, 2000; Yarar *et al*, 2005; Boucrot *et al*, 2006; Ferguson *et al*, 2009; Galletta and Cooper, 2009; Wu *et al*, 2010; Boulant *et al*, 2011; Taylor *et al*, 2011; Anitei and Hoflack, 2012). In support of a role of F-actin in CME it has been reported that BAR domain-containing proteins such as endophilin directly bind to the plasma membrane to sense or induce membrane curvature and cooperate with the actin cytoskeleton during membrane invagination (Yarar *et al*, 2005; Ferguson *et al*, 2009; Wu *et al*, 2010; Suetsugu and Gautreau, 2012) and scission (Itoh *et al*, 2005; Yarar *et al*, 2005; Tsujita *et al*, 2006).

Furthermore, branched F-actin structures, reminiscent of Arp2/3 nucleated branched arrays in lamellipodia, have been visualized at clathrin-coated pits (CCPs) (Collins *et al*, 2011). Fittingly, the Arp2/3 activator N-WASP contributes to epidermal growth factor receptor (EGFR) endocytosis (Kessels and Qualmann, 2002; Merrifield *et al*, 2004; Benesch *et al*, 2005). In lamellipodia, the length and branching of actin filaments are antagonistically regulated by the Arp2/3 complex and Ena/VASP proteins (Krause *et al*, 2003; Pula and Krause, 2008). Mena but not other Ena/VASP proteins have been implicated in EGF-dependent breast cancer invasion and metastasis, however, how Mena is linked to the EGFR is unknown (Philippar *et al*, 2008). Furthermore, Lamellipodin and proteins regulating elongation of actin filaments such as Mena have not been implicated in endocytosis.

Here, we show that Lpd forms protein complexes with endophilin and the EGFR and discovered direct interactions that link endophilin to Lpd-Ena/VASP. We provide the novel mechanistic insight that endophilin, Lpd, and Mena regulate EGFR endocytosis and that Lpd's function in this process requires Ena/VASP interaction and F-actin. Actin polymerization may support membrane invagination and scission during endocytosis. Here, we provide good evidence that Lpd is recruited to CCPs just before scission. We have identified a unique pathway in which Lamellipodin functions downstream of endophilin to regulate the F-actin cytoskeleton via Mena to support CCP scission during EGFR endocytosis.

## Results

Using total internal reflection fluorescence (TIRF) microscopy, we observed that EGFP-Lpd localizes not only to

protruding lamellipodia and filopodia (Krause *et al*, 2004) but also to rapidly disappearing spots at the plasma membrane reminiscent of CCPs (Supplementary Movie S1). To identify proteins that may link Lpd with regulators of endocytosis, we conducted a proteomic screen of a human fetal brain protein array with *in vitro* translated, $^{35}$S-labelled full length Lpd as the bait (Supplementary Figure S1A and B). Interestingly, several positive hits contained the SH3 domain of endophilin A1 (Endo1, SH3GL2) and endophilin A3 (Endo3, SH3GL3) but were N-terminally truncated. Members of the mammalian endophilin A family, which includes the additional isoform endophilin A2 (Endo2, SH3GL1), contain an N-terminal N-BAR domain with membrane curvature-generating/ sensing properties, a C-terminal SH3 domain (Supplementary Figure S1C), and have been implicated in the regulation of vesicle endocytosis (Ringstad *et al*, 1997; Schuske *et al*, 2003; Verstreken *et al*, 2003; Chang-Ileto *et al*, 2011; Milosevic *et al*, 2011).

SH3 domains bind to specific proline-rich peptides and there are several putative SH3-binding sites located in the C-terminus of Lpd and RIAM. To test whether the endophilin SH3 domain mediates the interaction with Lpd and whether it also binds RIAM, we performed pull-down assays from lysates of NIH/3T3 cells with purified GST-SH3 domains of each endophilin isoform. Lpd (Figure 1A) but not RIAM (Supplementary Figure S1D) was pulled down by all endophilin SH3 domains suggesting an SH3 domain-mediated

interaction of endophilin specifically with Lpd. Furthermore, co-immunoprecipitation of endogenous Lpd and endophilin A3 indicates that Lpd is indeed a novel binding partner of endophilin in mammalian cells (Figure 1B; see Supplementary Figure S1E for antibody specificity).

As described previously, overexpression of endophilin A3 induces membrane tubulation (Ferguson *et al*, 2009) (Figure 1E), while endophilins A1 and A2 localize to CCPs (Perera *et al*, 2006) (Figure 1C and D). We individually co-expressed the GFP-endophilin isoforms with mCherry-Lpd in HeLa cells and used TIRF microscopy to selectively analyse whether Lpd colocalizes with endophilin at the plasma membrane. We observed colocalization of Lpd with endophilin A1 (Figure 1C), A2 (Figure 1D), and A3 (Figure 1E) in 83%, 71%, and 93% of the cells, respectively (Figure 1F). As expected, the Lpd-related protein RIAM did not colocalize with endophilin (not shown).

To identify the part of Lpd that interacts with the SH3 domain of endophilin, we generated different truncation mutants of Lpd tagged with mCherry (Figure 2A) and co-expressed them with GFP-endophilin A3 in HeLa cells. We scored colocalization of both proteins in cells in which the expression of endophilin A3 caused membrane tubulation. As expected, we did not observe colocalization of the two C-terminal truncation mutants of Lpd (Lpd-N1, Lpd-N2) with endophilin A3, as these do not contain SH3 domain-binding sites (Figure 2B and C). However, all N-terminal

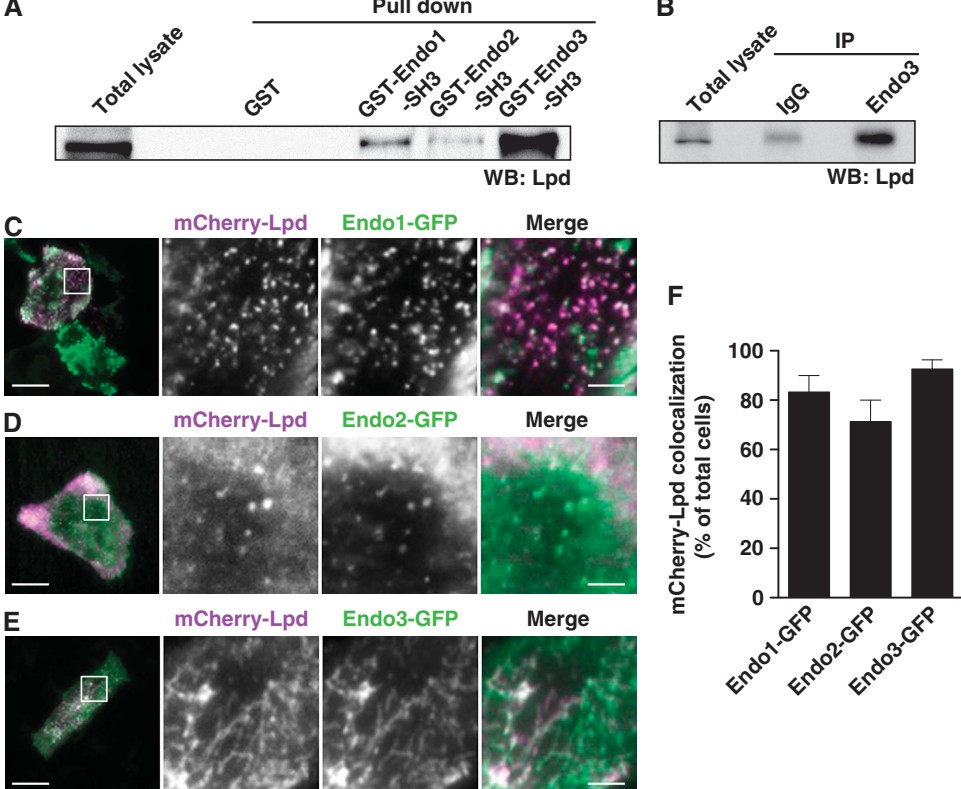

**Figure 1** Lamellipodin and endophilin interact in cells. (**A**) Pull down of Lpd from NIH/3T3 cell lysate using GST-tagged SH3 domains of endophilin A1 (Endo1), endophilin A2 (Endo2), and endophilin A3 (Endo3) or GST as a control. (**B**) IP of endophilin A3 from NIH/3T3 cell lysate using Endo3-specific antibodies or control IgG. (**A**, **B**) The western blots were probed with anti-Lpd antibodies. A representative blot from three independent experiments is shown. (**C–E**) HeLa cells expressing mCherry-Lpd and (**C**) Endo1-GFP, (**D**) Endo2-GFP, or (**E**) Endo3-GFP were imaged using TIRFM. Single colour (magnified square) and merged images of a representative cell are shown. Scale bar: 30 μm (left image) and 5 μm (right image). (**F**) mCherry-Lpd colocalization with Endo1-GFP, Endo2-GFP, and Endo3-GFP was scored in at least 30 cells each from 3 independent experiments. Lpd was considered to colocalize when it overlapped with the majority of endophilin spots/tubules.

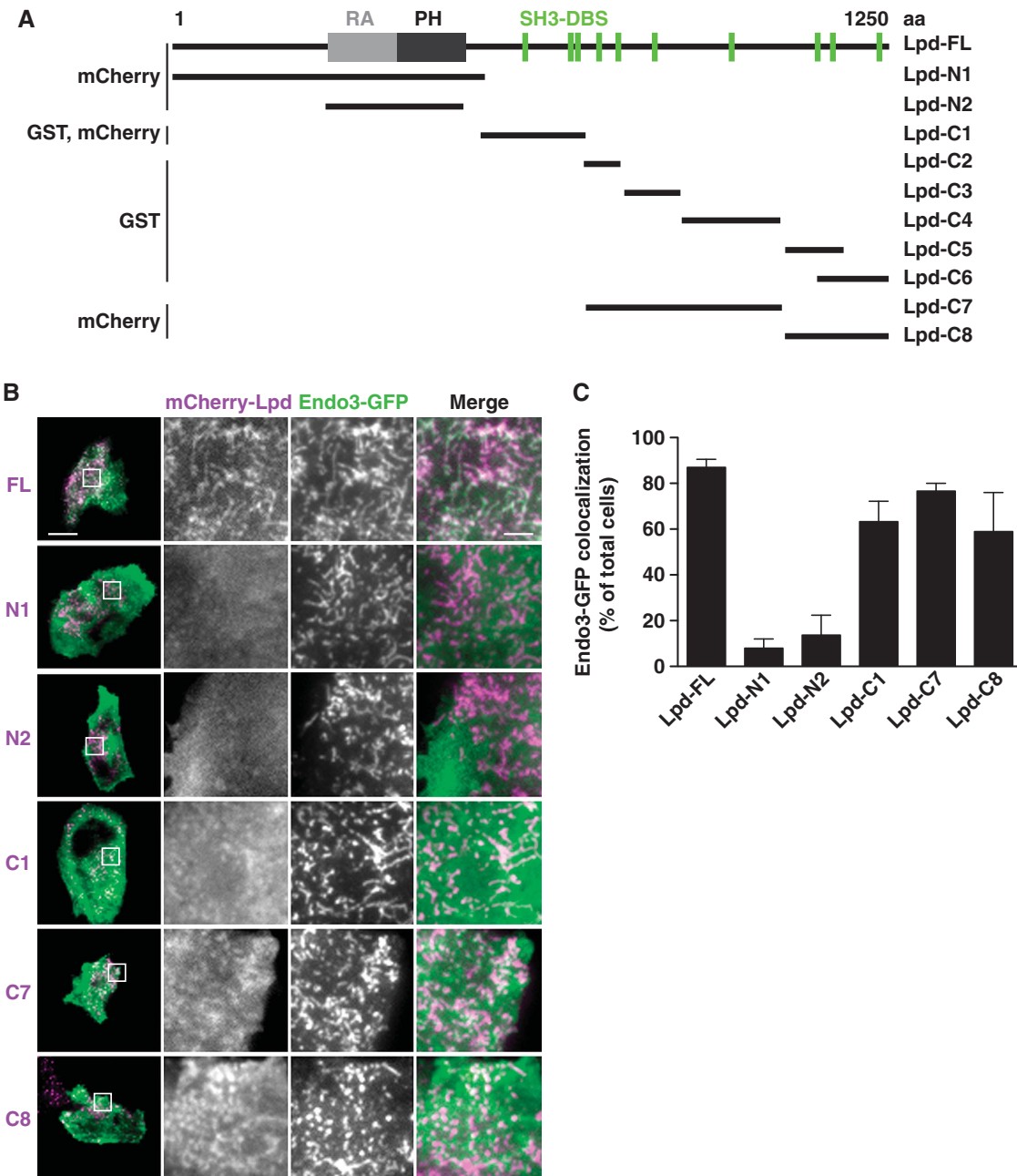

**Figure 2** Endophilin colocalizes with the C-terminus of Lamellipodin at induced membrane tubules. (**A**) Full-length (FL) and truncation mutants of Lpd with their respective protein tags used in this study are shown. Location of the Ras association (RA) and Pleckstrin homology (PH) domain and SH3 domain-binding sites (SH3-DBS) are indicated. (**B**) GFP-Endo3 colocalization with different mCherry-Lpd truncation mutants (see **A**) overexpressed in HeLa cells. Images were acquired by TIRFM. A representative cell is shown. Scale bar: 30 μm (left image) and 5 μm (right image). (**C**) Endo3-GFP colocalization with different mCherry-Lpd truncation mutants (see **B**) was scored in at least 25 cells from 3 independent experiments. Lpd was considered to colocalize when it overlapped with the majority of Endo3 tubules.

truncated Lpd constructs containing proline-rich regions co-localized with endophilin A3 (Figure 2B and C), suggesting that the endophilin SH3 domain binds to various proline-rich regions in the Lpd sequence. To investigate whether Lpd and endophilin interact directly, fragments of Lpd covering the whole C-terminus were fused to GST (Figure 2A). In a Far Western assay, the purified GST-Lpd fusion proteins were overlaid with the purified SH3 domain of each endophilin isoform fused to maltose-binding protein (MBP). In agreement with our colocalization data, the MBP-endophilin A1, A2, and A3-SH3 domains bound to all Lpd constructs

containing SH3 domain-binding sites (Figure 3A–C). MBP appeared to non-specifically bind to the highly charged C-terminal-most sequence of Lpd (Figure 3D), although it did not bind GST alone, which served as the negative control (Figure 3A–C). This indicates that all endophilin SH3 domains can interact directly with several proline-rich regions within Lpd.

To determine the specific endophilin SH3 domain-binding sites in Lpd, we designed a SPOTS scan peptide array with consecutive 12mer peptides that overlap each other by three amino acids and cover the complete C-terminus of Lpd.

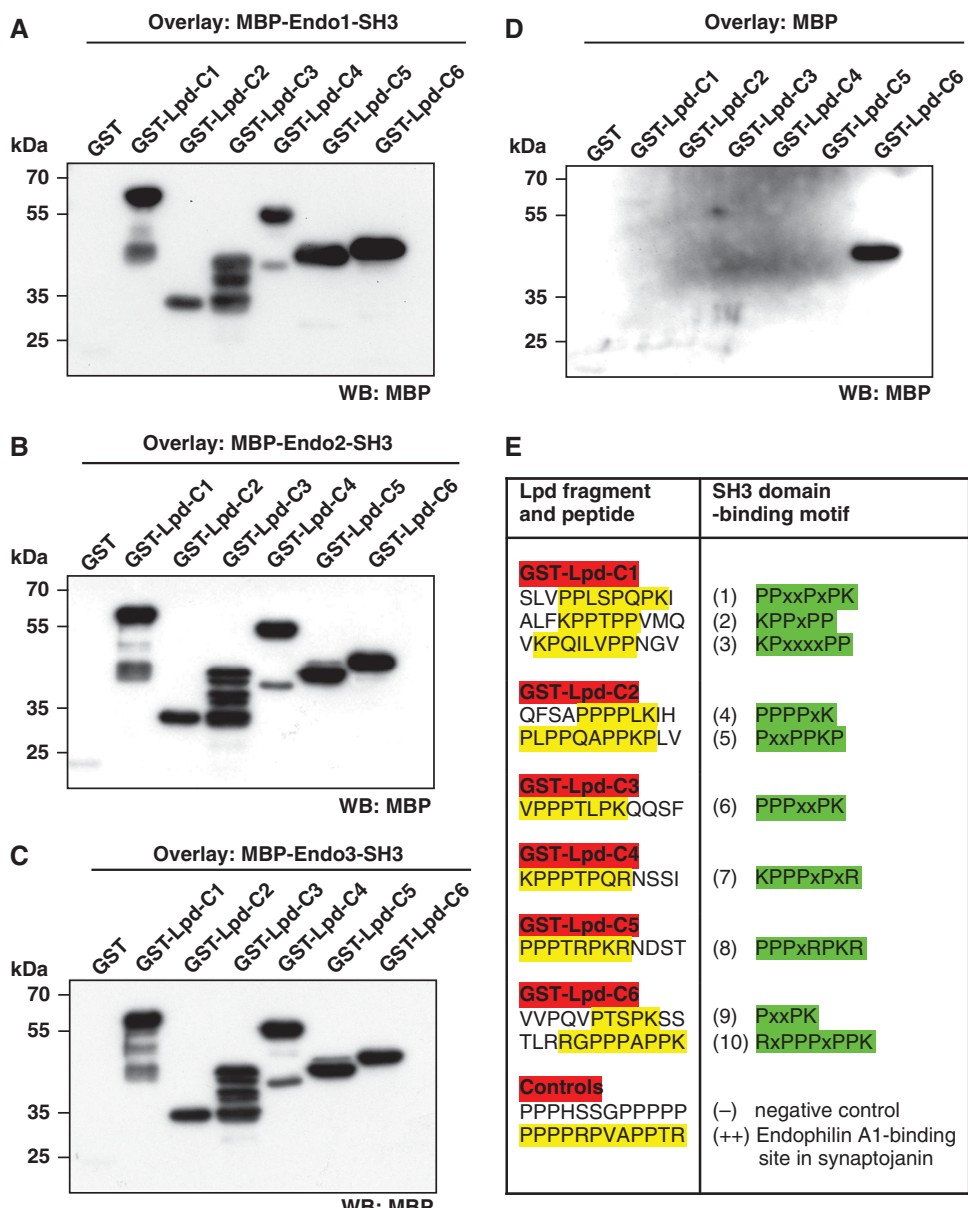

**Figure 3** The endophilin SH3 domain directly interacts with several SH3 domain-binding sites within Lamellipodin. (A–D) Far western overlay of different GST-Lpd truncation mutants (see Figure 2A) or GST control using (**A**) MBP-Endo1-SH3 domain, (**B**) MBP-Endo2-SH3 domain, (**C**) MBP-Endo3-SH3 domain, and (**D**) MBP only. A representative blot of three independent experiments is shown. (**E**) Table shows SH3 domain binding motifs in the Lpd sequence identified in the SPOTS scan peptide array overlaid with MBP-Endo2-SH3 domain. For detailed results, see Supplementary Figure S2.

We overlaid this SPOTS scan array with the purified MBP-SH3 domain of endophilin A2 that is the predominant isoform in non-neuronal cells (Ringstad *et al*, 1997). Detection of MBP revealed that endophilin A2 binds directly to 10 SH3 domain-binding sites within Lpd (Figure 3E; Supplementary Figure S2A and B) and suggests that Lpd might simultaneously bind several endophilin proteins, which assemble at invaginating membrane tubules during the highly organized endocytosis process.

Taken together, we found that the endophilin SH3 domain directly interacts with 10 potential SH3-binding sites in the C-terminus of Lpd, both proteins colocalize, and form a complex with each other in cells.

The EGFR is mainly internalized via CME upon exposure to physiological levels of EGF (2 ng/ml) (Lund *et al*, 1990;

Vieira *et al*, 1996; Huang *et al*, 2004; Sigismund *et al*, 2008). Endophilin has been suggested to function in EGFR endocytosis and is recruited to activated EGFR complexes (Soubeyran *et al*, 2002), but its role has not been tested directly. Because endogenous Lpd co-immunoprecipitates with endophilin (Figure 1B) and both colocalize at the plasma membrane (Figure 1C–E), we hypothesized that also Lpd might form a complex with the EGFR and localize to CCPs. TIRF microscopy revealed that mCherry-Lpd and EGFR-GFP do indeed colocalize at the plasma membrane in clusters resembling CCPs (Figure 4A). We tested whether Lpd interacts with the EGFR by overexpressing EGFR-GFP in HEK293 cells and immunoprecipitation of endogenous Lpd. Probing of the precipitates with anti-GFP antibodies showed a co-immunoprecipitation with the specific Lpd antibody but

not the control IgG (Figure 4B). We also observed co-immu-noprecipitation of endogenous Lpd and EGFR in A431 (Figure 4C and D) and in HeLa cells (Figure 4E and F), which suggests that the actin cytoskeletal regulator Lpd and the EGFR form a protein complex in cells.

In addition, we detected colocalization of GFP-Lpd-positive clusters with mRFP-clathrin light chain (Clc) at CCPs (Figure 4G). On average 28% of clathrin spots colocalized with Lpd, whereas 53% of Lpd-positive spots colocalized with clathrin (Figure 4G), suggesting that Lpd is dynamically recruited to CCPs. Since Lpd functions by recruiting the actin cytoskeleton regulatory Ena/VASP proteins (Krause *et al*, 2004; Michael *et al*, 2010), we tested whether GFP-Mena and GFP-VASP also colocalizes with mRFP-Clc at CCPs and we found that this is indeed the case (Figure 4H; Supplementary Figure S3A and B; Supplementary Movie S2). On average 29% of clathrin spots colocalized with Mena, whereas 31% of Mena spots colocalized with clathrin (Figure 4H). To analyse whether Lpd dynamically localizes to CCPs, we quantified Lpd recruitment during CCP scission in TIRFM movies. This analysis revealed that Lpd-GFP colocalized with 57% of mRFP-Clc labelled CCPs shortly before their scission (Figure 4I and J; Supplementary Movie S3). These data led us to hypothesize that Lpd and Ena/VASP proteins may regulate the actin cytoskeleton at CCPs to support CME of the EGFR.

However, the role of the actin cytoskeleton in CME is controversial and whether actin polymerization is required for EGFR uptake is unknown. To test this, we used an ELISA-based EGFR internalization assay in HeLa cells (Figure 5A). We assessed the percentage of EGFR uptake in cells after F-actin depolymerization with the G-actin sequestering drug Latrunculin B (Lat B) or with DMSO as a control. We observed that blocking F-actin polymerization upon physio-logical EGF stimulation decreased the uptake of the EGFR by 20% (Figure 5B) and 38% at higher EGF concentrations (Supplementary Figure S3C).

Since Lpd regulates the actin cytoskeleton by recruiting Ena/VASP proteins (Krause *et al*, 2004; Michael *et al*, 2010), we hypothesized that Lpd and Ena/VASP might regulate EGFR endocytosis. We first assessed the effect of Lpd over-expression on EGFR endocytosis. Surprisingly, overexpression of Lpd-GFP substantially increased EGFR endocytosis by 51% at 2 ng/ml (Figure 5C) and 27% at 100 ng/ml EGF (Supplementary Figure S3D) compared to a GFP-only control. To verify the function of Lpd for EGFR internalization, we

efficiently knocked down Lpd expression with three Lpd-specific shRNAs (Supplementary Figure S3E). We observed a non-significant reduction in EGFR endocytosis by ~56% after 2 min stimulation with 2 ng/ml EGF in Lpd knock-down cells (Supplementary Figure S3F). After 5 or 20 min stimulation, EGFR endocytosis is significantly decreased by ~25% at both 2 ng/ml (Figure 5D and E) and 100 ng/ml (Supplementary Figure S3G) of EGF.

CME of the EGFR only occurs after EGF stimulation, however, clathrin-independent mechanisms may also ac-count for some of the internalized EGFRs under certain circumstances (Lund *et al*, 1990; Yamazaki *et al*, 2002; Sigismund *et al*, 2005; Orth *et al*, 2006) and many other receptors including the transferrin receptor are taken up by constitutive CME mechanisms (Warren *et al*, 1997; Johannessen *et al*, 2006). Fluorescently labelled transferrin uptake was not affected by Lpd knockdown in HeLa cells in an imaging-based assay (Supplementary Figure S4A and B).

To explore the function of Ena/VASP proteins in CME of the EGFR, we efficiently knocked down Mena or VASP expression with two independent Mena or VASP-specific shRNAs, re-spectively (Supplementary Figure S5A). Interestingly, knock-down of Mena but not VASP decreased EGFR endocytosis by up to 58% after 5, 10, and 15 min stimulation with 2 ng/ml EGF (Figure 5F and G; Supplementary Figure S5B and C).

Thus, these data show that Lpd, Mena, and F-actin con-tribute specifically to CME of the EGFR (Figure 5B–G), suggesting that Lpd may regulate EGFR endocytosis via Mena and F-actin.

To test this further, we overexpressed Lpd-GFP or GFP with or without simultaneous addition of Lat B. Lamellipodin overexpression did not increase EGFR endocytosis when actin polymerization was inhibited (Figure 6A and B), indicating that Lpd indeed regulates EGFR endocytosis via F-actin.

Since Lpd regulates the actin cytoskeleton via Ena/VASP proteins, Mena localizes at CCPs, and Mena is required for EGFR endocytosis (Figures 4H, 5F, and G), we also investi-gated whether the function of Lpd for EGFR internalization depends on its interaction with Mena. In contrast to Lpd-GFP, Lpd-F/A-GFP, a mutant of Lpd in which all seven Ena/VASP-binding sites had been mutated (Krause *et al*, 2004), did not increase EGFR endocytosis (Figure 6C and D). Taken together, our data identify an important role of F-actin polymerization for EGFR uptake and suggest that Lpd regulates F-actin-dependent endocytosis of the EGFR via Mena.

**Figure 4** Lamellipodin is recruited to CCPs and interacts with the EGFR. (**A**) HeLa cells expressing mCherry-Lpd and EGFR-GFP were imaged using TIRFM. Single colour (magnified square) and merged images of one representative cell are shown. Scale bar: 30 µm (left image) and 5 µm (right image). (**B**) IP of EGFR from HEK-293 cells overexpressing EGFR-GFP using Lpd-specific antibodies or IgG control. EGFR was detected using anti-GFP antibodies (left panels). Reprobe of the same blot with Lpd-specific antibodies (right panels). (**C, D**) Co-IP of endogenous EGFR and Lpd from A431 cell lysate using Lpd (**C**) or EGFR-specific antibodies (**D**) or IgG control. EGFR and Lpd were detected using specific antibodies. (**E, F**) Co-IP of endogenous EGFR and Lpd from HeLa cell lysate using Lpd (**E**) and EGFR-specific antibodies (**F**) or IgG control. Cells were stimulated with 2 ng/ml EGF ( + ) for 5 min or not stimulated ( − ). (**B–F**) A representative blot each from at least three independent experiments is shown. (**G, H**) HeLa cells expressing mCherry-Lpd and GFP-Lpd or (**H**) GFP-Mena and mRFP-Clc were imaged using TIRFM. Single colour (magnified square) and merged images of one representative cell are shown. (**G, H**) Scale bar: 30 µm (**G**) and 10 µm (**H**) (left image) and 5 µm (**G**) and 2 µm (**H**) (right image). (**G, H**) Quantification of the percentage of colocalization of mRFP-Clc with Lpd-GFP (Clathrin versus Lpd) (**G**) or GFP-Mena (Clathrin versus Mena) (**H**) and Lpd-GFP with mRFP-Clc (Lpd versus Clathrin) (**G**) or GFP-Mena (Mena versus Clathrin) (**H**). Each time point of TIRF movies from four cells were analysed containing on average 850 Clc-positive and 450 Lpd-positive spots each. (**I, J**) Dynamics of Lpd-GFP and mRFP-Clc in HeLa cells was assessed every 5 s using TIRFM. Single colour and merged images of an area of a representative cell are shown. Arrows show recruitment of Lpd-GFP to mRFP-Clc shortly before scission. Scale bar: 1 µm (see also Supplementary Movie S3). (**J**) Quantification of the percentage of scission events of CCPs containing mRFP-Clc and Lpd-GFP. In total, 700 scission events of 3 different cells were analysed for each experiment.

Both endophilin and Lpd form protein complexes with the EGFR (Figure 4B–F) (Soubeyran *et al*, 2002) and Lpd regulates EGFR endocytosis (Figure 5D and E; Supplementary Figure S3E–G). Therefore, we hypothesized that endophilin regulates EGFR internalization via Lpd. To test this, we first assessed the percentage of EGFR uptake in HeLa cells over-expressing Endo3-GFP or GFP as a control. We observed that overexpression of Endo3-GFP significantly increased EGFR uptake by ~40% at physiological and high concentrations of EGF (Figure 6E and F). This is a similar increase in EGFR

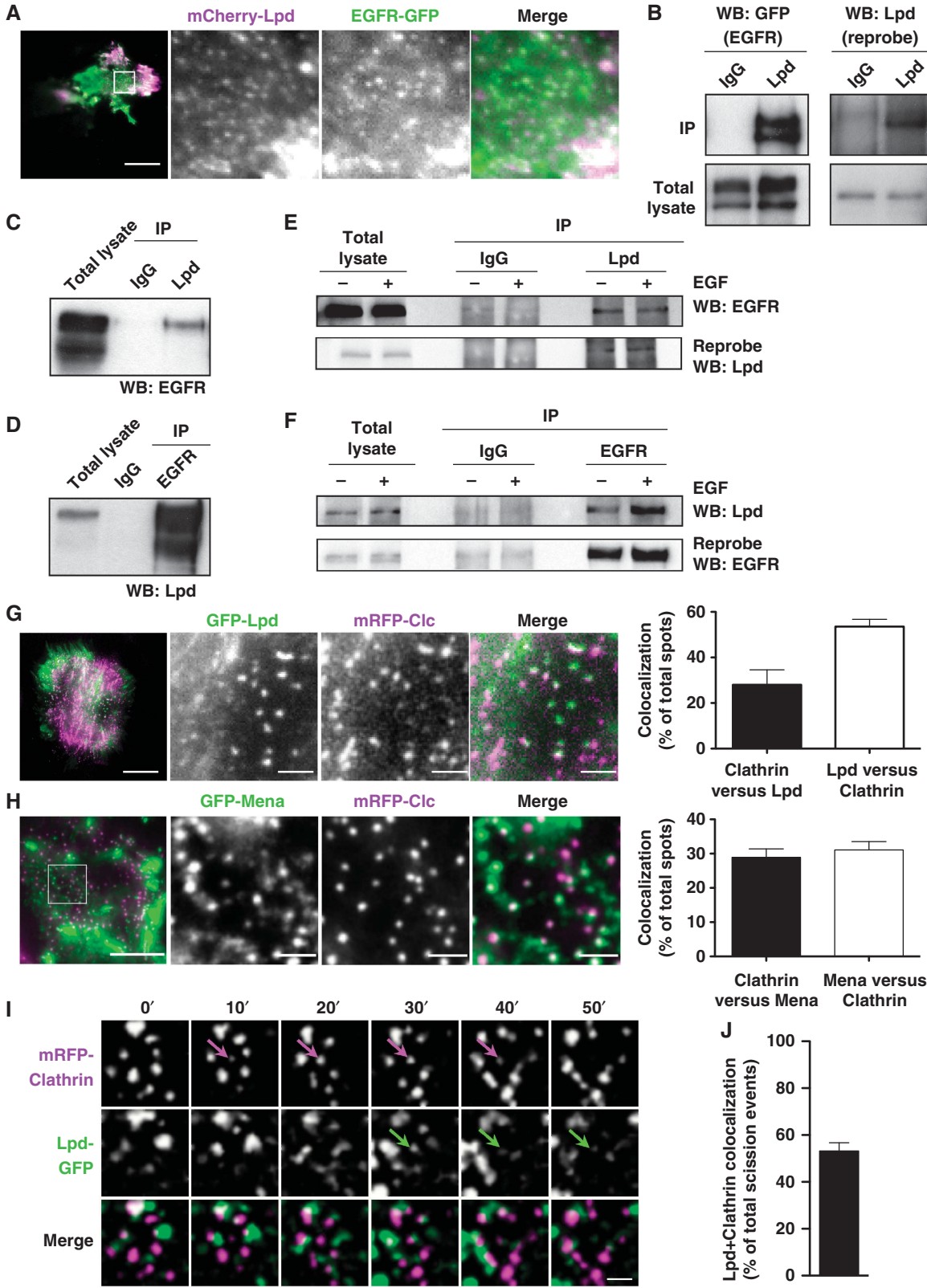

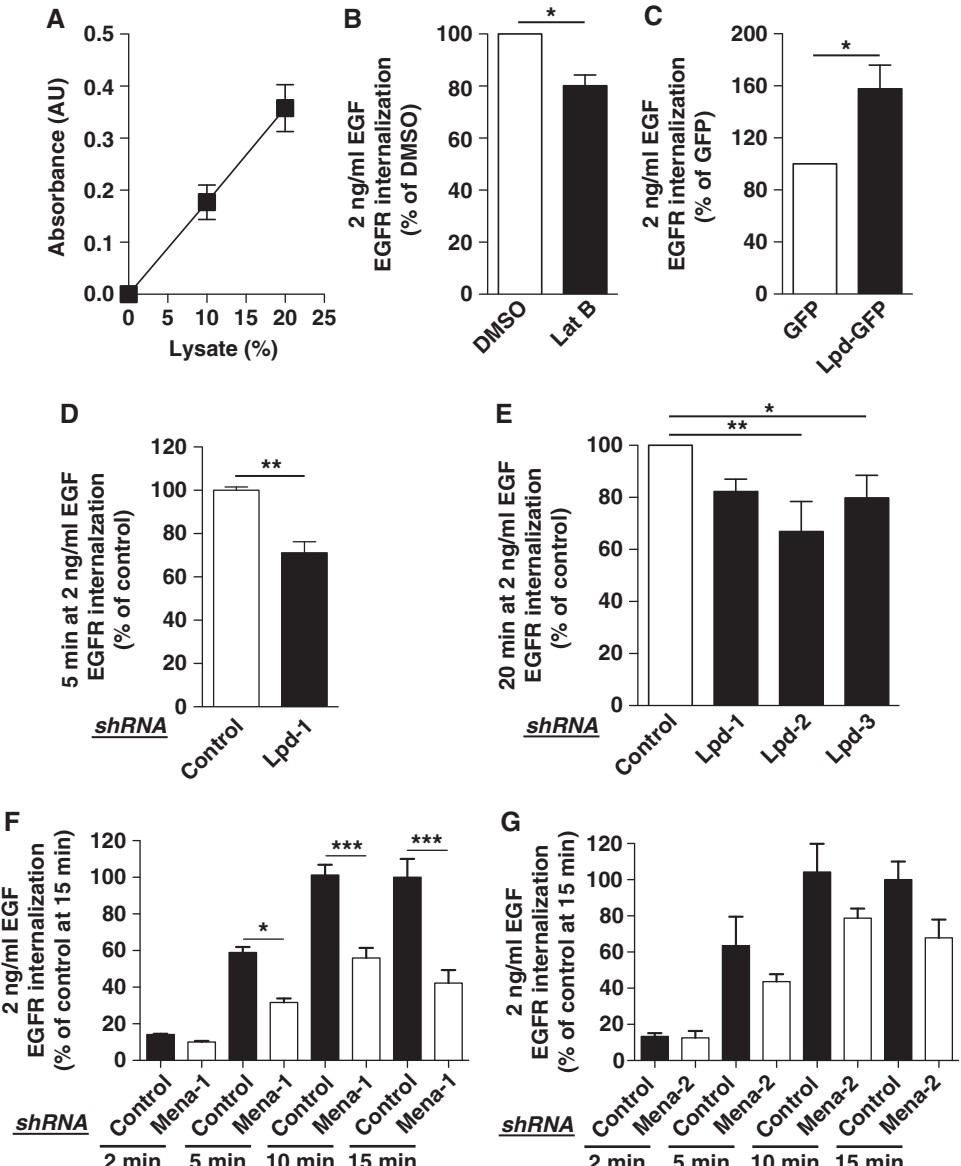

**Figure 5** Lamellipodin, Mena, and F-actin regulate EGFR internalization. (**A**) Linear increase in the absorbance of biotinylated surface EGFR with increasing amounts of lysates of HeLa cells. Values are mean ( ± s.e.m.) of six independent experiments. (**B**) EGFR internalization in HeLa cells treated with Latrunculin B (Lat B) or DMSO control and 2 ng/ml EGF. (**C**) EGFR internalization in HeLa overexpressing Lpd-GFP or GFP as a control and treated with 2 ng/ml EGF. (**D–G**) EGFR internalization in HeLa cells expressing three Lpd-specific (**D**, **E**) or two Mena-specific (**F**, **G**) or control shRNA and treated with 2 ng/ml EGF for indicated times. (**B–G**) Results are mean ± s.e.m. of at least three independent experiments. (**B–D**) t-test: *$P<0.05$, **$P<0.01$. (**E–G**) One-way ANOVA, Tukey's: *$P<0.05$, **$P<0.01$, ***$P<0.001$.

uptake as induced by overexpression of Lpd-GFP (Figure 5C Supplementary Figure S3D). We then examined whether the function of endophilin in EGFR endocytosis is mediated via Lpd by overexpressing GFP-endophilin A3 or GFP in combination with Lpd-specific or control shRNA. Overexpression of GFP-endophilin A3 significantly increased EGFR uptake in the presence of the control shRNA but not when the Lpd-specific shRNA was expressed (Figure 6E and F), suggesting that endophilin functions in EGFR endocytosis and this is mediated by Lpd.

Since Lpd functions downstream of endophilin and links the actin cytoskeleton with EGFR endocytosis, we explored whether Lpd regulates the actin cytoskeleton to support membrane invagination or scission using mouse embryonic fibroblasts lacking all dynamin isoforms (DKO = dynamin1/2 KO MEFs) (Ferguson et al, 2009). In the absence of dynamin, membrane scission is reduced, resulting in an accumulation of arrested CCPs with long tubular necks that contain endophilin, N-WASP, α-adaptin, and F-actin (Figure 7A and B, and not shown) (Ferguson et al, 2009). We observed that endogenous Lpd (Figure 7C) and Mena (Figure 7D) also colocalize with these F-actin clusters at CCPs. Treatment of these cells with Lat B results in the conversion of long tubular necks into short, wide necks, suggesting that actin polymerization supports both membrane invagination and scission (Ferguson et al, 2009). We reasoned that the absence of a protein that regulates the actin cytoskeleton to drive membrane invagination would decrease the density

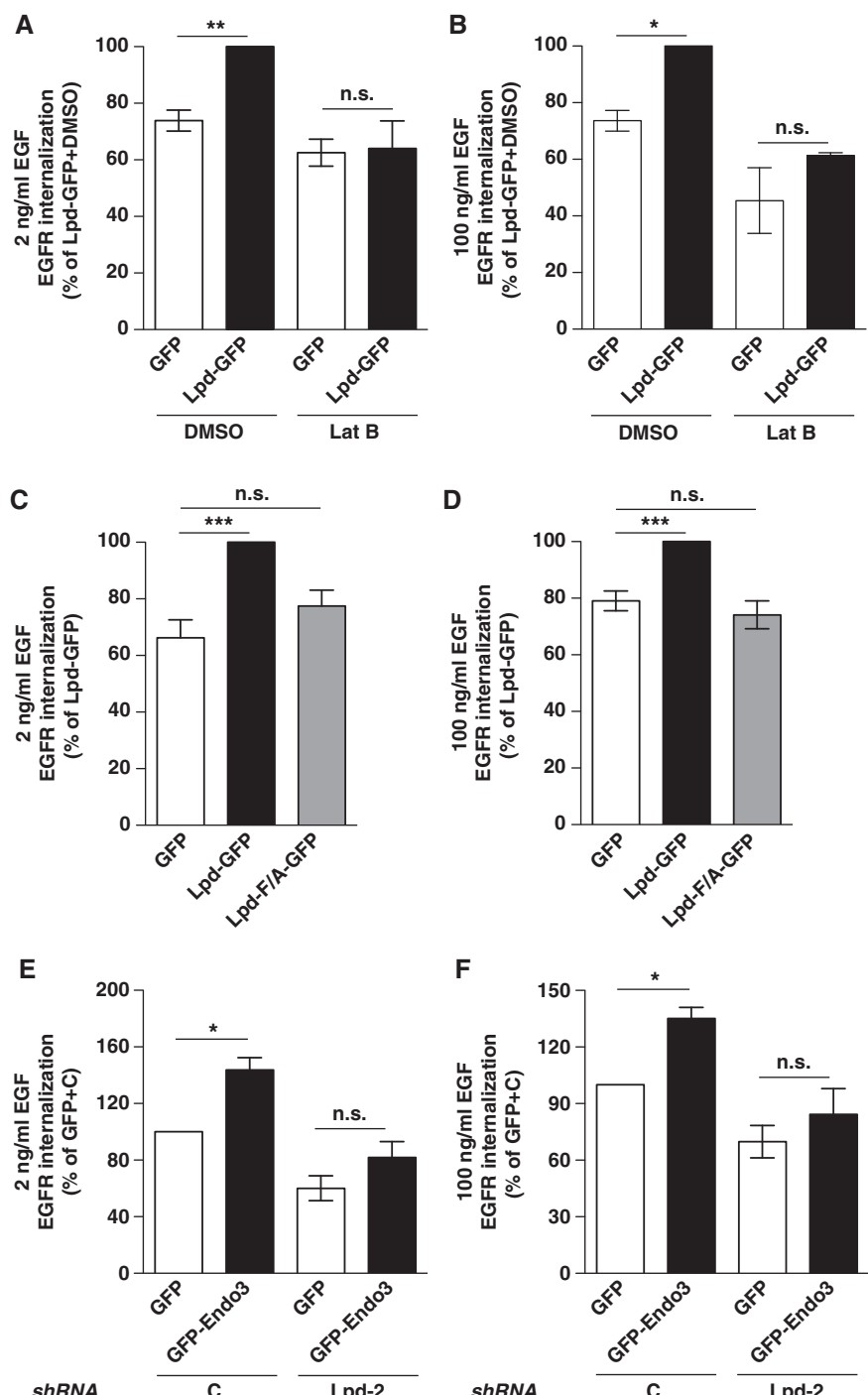

**Figure 6** Endophilin and Lamellipodin cooperate to regulate actin-dependent EGFR internalization via Ena/VASP. (**A–F**) EGFR internalization in HeLa cells (**A**, **B**) overexpressing Lpd-GFP or GFP as control and treated with Lat B or DMSO control, (**C**, **D**) overexpressing Lpd-GFP, Lpd-F/ A-GFP or GFP as a control, and (**E**, **F**) overexpressing GFP-Endo3 or GFP as a control and Lpd-specific or control shRNA. (**A**, **C**, **E**) Cells were stimulated with 2 ng/ml EGF. (**B**, **D**, **F**) Cells were stimulated with 100 ng/ml EGF. (**A–F**) Results are mean ± s.e.m. of at least three independent experiments. One-way ANOVA, Tukey's: *$P<0.05$, **$P<0.01$, ***$P<0.001$, n.s. not significant.

of arrested CCPs, whereas knockdown of a protein that regulates the actin cytoskeleton to drive scission would increase the density of arrested CCPs. Interestingly, knockdown of Lpd in the DKO MEFs significantly increased the number of arrested CCPs per $\mu m^2$, indicating that Lpd regulates the actin cytoskeleton to support vesicle scission (Figure 7E and F). To further explore the role of Lpd in CCP

scission during EGFR endocytosis, we examined Lpd knockdown and control HeLa cells after starvation and stimulation with 2 ng/ml EGF for 2 min by transmission electron microscopy. We observed more invaginated, omega-shaped, and tubulated CCPs in Lpd knockdown cells compared to control cells (Figure 7G) further supporting a role for Lpd in CCP scission.

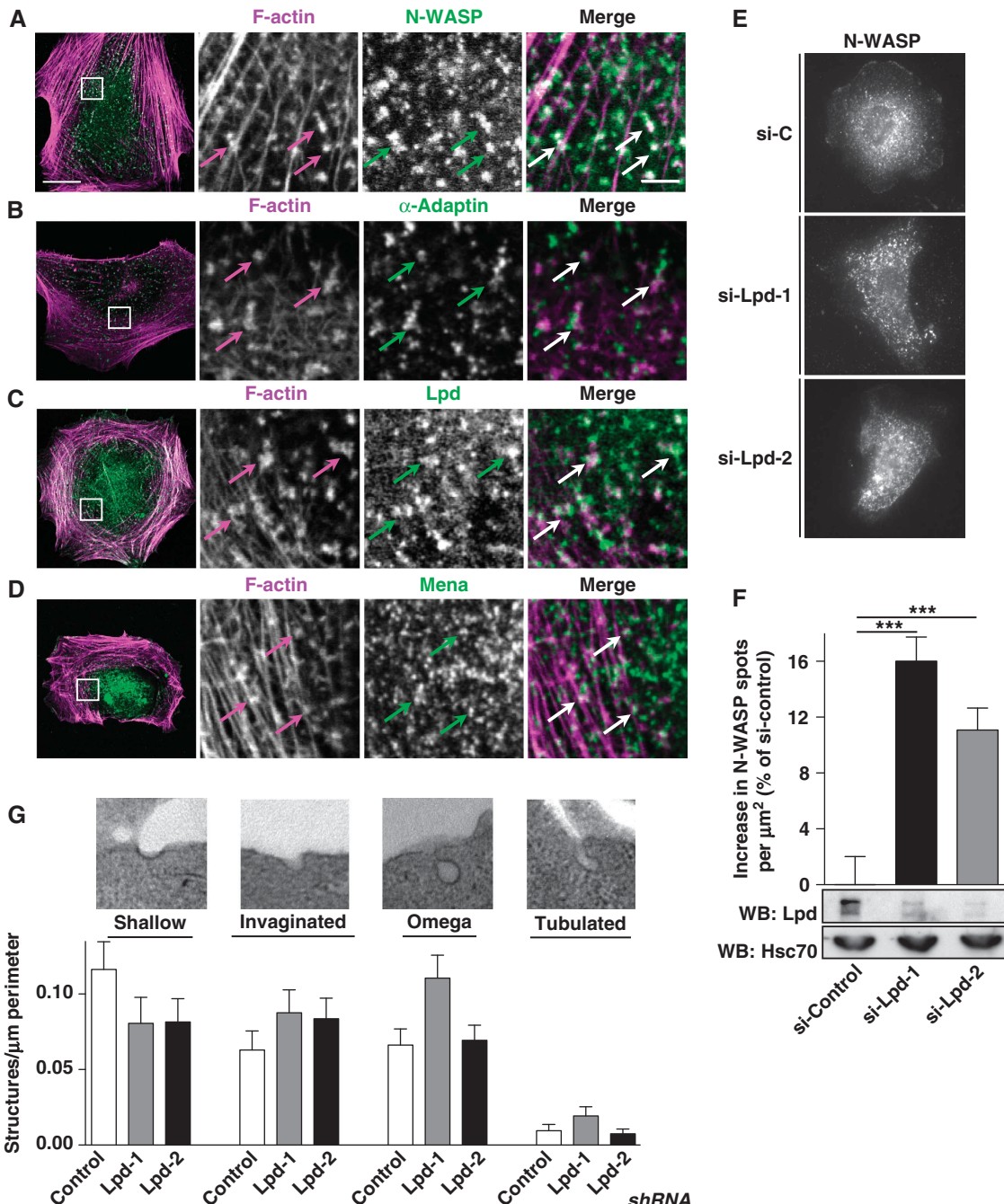

**Figure 7** Lamellipodin is implicated in CCP scission. (**A–D**) IF staining in DKO MEFs for (**A**) N-WASP and F-actin, (**B**) α-adaptin and F-actin, (**C**) Lpd and F-actin, and (**D**) Mena and F-actin. (**A–D**) Cells were imaged using confocal microscopy. Single colour (magnified square) and merged images of a representative cell are shown. Scale bar: 20 μm (left image) and 2.5 μm (right image). (**E**) IF staining of N-WASP in DKO MEFs treated with two different Lpd-specific (si-Lpd1, si-Lpd-2) or control siRNA (si-C). (**F**) Quantification of the number of N-WASP spots per μm$^2$ in DKO MEFs from (**E**), lower panels show lysates of DKO MEFs treated with si-C, si-Lpd-1, and si-Lpd-2. Lpd and Hsc70 (loading control) expression was detected using specific antibodies. One-Way ANOVA, Dunnett's, *** = $P < 0.001$. (**G**) Stages of CME were analysed by transmission electron microscopy in control shRNA expressing and Lpd knockdown HeLa cells after starvation and stimulation with 2 ng/ml EGF for 2 min. Shallow, invaginated, omega-shaped, and tubulated CCPs were scored by blinded observers from 20 cells for each control and knockdown from two independent experiments.

Taken together, our data suggest that downstream of endophilin, Lpd and Mena regulate the actin cytoskeleton to support membrane scission during CME of the EGFR.

## Discussion

In this study, we have identified Lpd as a novel binding partner of the endophilin SH3 domain and a mediator of endophilin's function. Endophilin is recruited together with Arp2/3 and dynamin at late stages of CCP invagination just before scission (Perera *et al*, 2006; Taylor *et al*, 2011). The N-BAR domain senses narrow tube diameters similar to the neck of a clathrin-coated bud and is sufficient for the recruitment of endophilins to these sites (Milosevic *et al*, 2011). The N-BAR domain is composed of an N-terminal amphipathic helix, which supports membrane scission via a

shallow hydrophobic insertion into the plasma membrane, and a BAR domain that can stabilize or induce a narrow neck *in vitro* (Farsad *et al*, 2001; Gallop *et al*, 2006; Masuda *et al*, 2006; Boucrot *et al*, 2012; Mim *et al*, 2012). The SH3 domain of endophilin A2 directly interacts with 10 proline-rich peptides that are widely distributed throughout the C-terminus of Lpd. The motifs that are recognized by the endophilin A2 SH3 domain have an unconventional consensus sequence, in agreement with phage display and peptide mutagenesis results from the known endophilin A2 SH3-binding sites in synaptojanin and dynamin (Cestra *et al*, 1999). Taken together, this suggests that Lpd is recruited by endophilin to CCPs.

What is the function of Lpd downstream of endophilin? In yeast, the endophilin orthologue Rvs167 is required for vesicle scission (Kaksonen *et al*, 2005). Endophilin recruits synaptojanin to facilitate membrane fission and vesicle uncoating (Schuske *et al*, 2003; Verstreken *et al*, 2003; Chang-Ileto *et al*, 2011; Milosevic *et al*, 2011). Furthermore, endophilin recruits dynamin and thereby supports membrane scission (Ringstad *et al*, 1997). Endophilin had also been observed before at 50–60% of CCPs just before scission (Perera *et al*, 2006; Taylor *et al*, 2011). We detected Lpd at a much higher percentage of scission events than other regulators of F-actin such as N-WASP, which localized only to 20% of scission events (Taylor *et al*, 2011). The actin cytoskeleton is essential for endocytosis in yeast but its role in mammalian CME is controversial. However, good evidence has been presented that actin polymerization promotes membrane invagination as well as scission (Galletta and Cooper, 2009; Anitei and Hoflack, 2012). Recently, a branched actin network, reminiscent of the actin network in lamellipodia, has been observed at CCPs supporting the hypothesis that actin polymerization plays a role in CME (Collins *et al*, 2011). However, it is unknown whether actin polymerization supports EGFR uptake. We show that 25% of clathrin-mediated EGFR endocytosis depends on F-actin polymerization in the presence of dynamin. The actin ultrastructure in lamellipodia is controlled by a balance of Arp2/3 activity increasing branching and Lpd-Ena/VASP activity increasing the length of actin filaments (Chesarone and Goode, 2009). The Arp2/3 activator N-WASP contributes to EGFR endocytosis (Kessels and Qualmann, 2002; Merrifield *et al*, 2004; Benesch *et al*, 2005; Innocenti *et al*, 2005). However, neither Lpd nor Ena/VASP has thus far been implicated in endocytosis. We show that endophilin can directly interact with Lpd (Figures 1 and 2), and that Lpd mediates endophilin's function in this process in an Ena/VASP and F-actin-dependent manner. We observed a dramatic increase in EGFR uptake upon endophilin or Lpd overexpression, suggesting that endophilin-Lpd regulated F-actin polymerization plays an important role in endocytosis of the EGFR. Furthermore, knockdown of Lpd and Mena (Figure 5) or the absence of N-WASP (Benesch *et al*, 2005; Innocenti *et al*, 2005) results in inhibition of EGFR uptake that is comparable to inhibition of F-actin polymerization. This suggests that the function of N-WASP-Arp2/3 and Lpd-Ena/VASP is coordinated to regulate formation of a branched actin network to support endocytosis.

Our data suggest that specifically Mena but not VASP regulates EGFR endocytosis (Figure 5F and G; Supplementary Figure S5A–C). This is intriguing since

Mena but not other Ena/VASP proteins have been implicated in EGF-dependent breast cancer invasion and metastasis. However, how Mena is linked to the EGFR is unknown (Philippar *et al*, 2008), and our combined data suggest that Lpd is the missing link between the EGFR and Mena.

Recently, it has been reported that formation of $PI(3,4)P_2$ spatiotemporally controls the maturation of late-stage CCPs towards scission (Posor *et al*, 2013). Importantly, Lpd is one of the few proteins harbouring a PH domain that is specific for $PI(3,4)P_2$ (Krause *et al*, 2004). The increase in the number of arrested CCPs in the absence of $PI(3,4)P_2$, Lpd, and dynamin (Figure 7) (Posor *et al*, 2013; Ferguson *et al*, 2009) and the increase in invaginated, omega-shaped, and tubulated CCPs in the Lpd knockdown cells (Figure 7) together with endophilin's known role in regulating scission indicate that Lpd functions downstream of endophilin and $PI(3,4)P_2$ to support membrane scission.

In summary, our results imply that endophilin and Lpd cooperate to regulate the F-actin cytoskeleton via Mena to support vesicle scission during endocytosis of the EGFR.

## Materials and methods

### Molecular biology

GFP-Lpd (Krause *et al*, 2004), mCherry-Lpd: full-length human Lpd (AY494951) in pCDNA3.1-mCherry-DEST which was generated by subcloning mCherry cDNA and the Gateway destination cassette (Invitrogen) into pCDNA3.1 (Invitrogen). Lpd-GFP or LpdF/A-GFP: Phenylalanine to alanine mutation in all seven Ena/VASP-binding sites (Krause *et al*, 2004) were introduced by site-directed mutagenesis (Quickchange, Agilent) into full-length human Lpd (AY494951) in pENTR3C (Invitrogen). Lpd and LpdF/A were transferred to pCAG-DEST-GFP, which was generated by subcloning the Gateway destination cassette (Invitrogen) into pCAG-GFP (Matsuda and Cepko, 2004) (kind gift of C Cepko, Harvard Medical School, Cambridge, USA; Addgene 11150). pBS-Lpd: full-length human Lpd (AY494951) in pBlueScript-II-SK(+) (Agilent). mRFP-Clc (Tagawa *et al*, 2005) (Addgene 14435) and EGFR-GFP were a kind gift from Dr Ari Helenius (ETH Zuerich) and Dr A Sorkin (University of Pittsburgh), respectively. pMSCV-GFP-Mena (Bear *et al*, 2000); pEGFP-VASP (mouse VASP cDNA cloned into *Bam*HI-*Eco*RI of pEGFP-C1; Clontech); Endophilin A1-GFP: human endophilin A1 in pDONR201 (HSCD00000899, Harvard Institute of Proteomics) was transferred into pDEST-EGFP, which was generated by subcloning the destination cassette (Invitrogen) into pEGFP-N1 (Clontech). Rat full-length endophilin A2-GFP was a kind gift from Dr Pietro De Camilli (Yale University, USA). Endophilin A3-GFP: the endophilin A3 cDNA (IMAGE5197246-AK68-m23) was cloned into pENTR11 (Invitrogen) and transferred to pDEST-EGFP. MBP or GST-endophilin A1-SH3, endophilin A2-SH3, endophilin A3-SH3: SH3 domain of endophilin A 1–3 in pMAL-c2g (New England Biolabs) or pGEX-6P1 (GE-Healthcare). mCherry-Lpd truncation constructs: amino acid (aa) numbering according to GenBank AAS82582 Lpd-N1 (aa1–592), Lpd-N2 (aa242–592), Lpd-C1 (aa593–727), Lpd-C7 (aa724–1092), Lpd-C8 (aa1093–1250) were cloned into pENTR3C and transferred to pCDNA3.1-mCherry-DEST by Gateway recombination. GST-Lpd truncation constructs: Lpd-C1 (aa545–728), Lpd-C2 (aa727–791), Lpd-C3 (aa791–889), Lpd-C4 (aa890–1062), Lpd-C5 (aa1030–1132), Lpd-C6 (aa1125–1250) were cloned into pGEX-6P1 (GE-Healthcare). Human Lpd, VASP, and Mena-specific shRNA and scrambled control shRNA constructs in pLL3.7puro (Rubinson *et al*, 2003): Lpd-1 (forward: 5′-tgcgtca aatcacagaaacgTTCAAGAGAcgtttctgtgatttgacgcTTTTTGGAAAGAATT CG-3′; reverse: 5′-tcgaCGAATTCTTTCCAAAAAgcgtcaaatcacagaaacg TCTCTTGAAcgtttctgtgatttgacgca-3′); Lpd-2 (forward: 5′-tgctctgaat cagggagagattcaagagatctctccctgattcagagcttttttggaaagaattcg-3′; reverse: 5′-tcgacgaattctttccaaaaagctctgaatcagggagagatctcttgaatctctccctgattca gagca-3′); Lpd-3 (forward: 5′-tgaacaggcctctttgagtaTTCAAGAGAtactc aaagaggcctgttctttttggaaagaattcg-3′; reverse: 5′-tcgacgaattctttccaaa aagaacaggcctctttgagtaTCTCTTGAAtactcaaagaggcctgttca-3′); Mena-1 (forward: 5′-tGCAGCAAGTCACCTGTTATCTCGAAAGATAACAGGT

GACTTGCTGCttttttggaaagaattcg-3′; reverse: 5′-TCGAcgaattcttccaa aaaGCAGCAAGTCACCTGTTATCTTTCGAGATAACAGGTGACTTGCT GCa-3′); Mena-2 (forward: 5′-TcgacaagagcagttagaaaTTCAAGAGAttt ctaactgctcttgtcgTTTTTTggaaagaattcg-3′; reverse: 5′-TCGAcgaattctt tccAAAAAcgacaagagcagttagaaaTCTCTTGAAtttctaactgctcttgtcgA-3′); VASP-1 (forward: 5′-tgagccaaactcaggaaagtTTCAAGAGAactttcctgagtt tggctcTTTTTTGGAAAGAATTCG-3′; reverse: 5′-tcgaCGAATTCTTT CCAAAAAgagccaaactcaggaaagtTCTCTTGAAactttcctgagtttggctca-3′; VASP-2 (forward: 5′-tgcgtccagatctaccacaattcaagagattgtggtagatctggac gctttttggaaagaattcg-3′; reverse: 5′-TCGAcgaattctttccaaaaagcgtccagatc taccacaatctcttgaattgtggtagatctggacgca-3′). All constructs were veri- fied by sequencing. Lpd ON-TARGETplus siRNA's (J-043405-11, J-043405-12) and control siRNA No 2 (Dharmacon).

## Antibodies

Anti-Lpd pab 3917 (Krause *et al*, 2004), anti-RIAM pab 4612 (Lafuente *et al*, 2004), anti-Mena mab A351F7D9 (Lebrand *et al*, 2004), anti-VASP mab (IE273; Immunoglobe, Germany; Niebuhr *et al*, 1997), anti-EGFR mab (Cancer Research UK), IP-specific anti- EGFR mab (Cell Signaling), anti-alpha-adaptin mab clone AP6 (Thermo Scientific), anti-N-WASP rabbit mab clone 30D10 (Cell Signaling); anti-endophilin A3 pab (S-15; Santa Cruz), anti-Hsc70 mab (B-6; Santa Cruz), anti-MBP mab (New England Biolabs); anti- GAPDH mab (clone 6C5; Millipore); anti-GFP mab (Roche).

## Cell culture and transfections

NIH/3T3, HEK293 cells (ATCC) and DKO MEFs (Ferguson *et al*, 2009) were grown in Dulbecco's Modified Eagle Medium (DMEM), 10% fetal bovine serum (FBS) and 2 mM L-glutamine. HeLa cells (ATCC) were grown in Minimum Essential Medium Eagle (MEM), 10% FBS, 1 × Non-essential amino acids and 1 mM sodium pyruvate. Cells were transfected using either Fugene HD or X-tremeGENE 9 (Roche). HeLa cells expressing control-, Mena, VASP, or Lpd-specific shRNAs were subjected to 3 days selection with 1 μg/ml Puromycin. Deletion of dynamin proteins in DKO MEFs was induced by the addition of 4-hydroxy-tamoxifen to the culture medium as described previously (Ferguson *et al*, 2009).

## Pull down, immunoprecipitation, and western blotting

Lysates were prepared in lysis buffer (50 mM Tris–HCl, pH 7.4, 200 mM NaCl, 1% NP-40, 2 mM $MgCl_2$, 10% glycerol, NaF, $Na_3VO_4$ and complete mini tablets without EDTA, Roche). Precleared lysate was incubated with glutathione beads (GE Healthcare) or with primary antibody or control IgG followed by protein A/G beads (Alpha Diagnostics) and was washed with lysis buffer. Western blotting was performed as described previously (Krause *et al*, 2004). Secondary antibodies were goat anti-rabbit, goat anti-mouse and rabbit anti-goat-HRP (Dako).

## EGF stimulation and EGFR internalization assay

EGFR internalization assay was done as described previously (Caswell *et al*, 2008) with slight variations. In brief: cells were plated at 50% confluence and starved overnight (MEM, 0.2% FBS). Cell-surface proteins were biotinylated with Sulfo-NHS-sulfo-SS- Biotin (Pierce) for 30 min at 4°C, washed with 1 × PBS and stimulated with 2 or 100 ng/ml EGF in starving medium for 2, 5, 10, 15, or 20 min. Remaining surface-bound biotin was removed by washes with reduction buffer (50 mM glutathione, 75 mM NaCl, 10 mM EDTA, 1% BSA, 75 mM NaOH) followed by washes with 5 mg/ml iodoacetamide. Amount of internalized EGFR was determined as the percentage of surface EGFR from the lysate of cells by ELISA. Internalized, biotinylated EGFR was captured by mouse-anti-human EGFR antibody (BD Pharmingen) and detected by HRP-coupled streptavidin (GE Healthcare).

## Immunofluorescence staining

Cells were plated on nitric acid-washed coverslips (Hecht-Assistant) and fixed in 4% paraformaldehyde-PHEM (60 mM PIPES, 25 mM HEPES, 10 mM EGTA, 2 mM $MgCl_2$, 0.12 M sucrose). Cells were permeabilized (1 × PBS, 0.05% Triton X-100) for 10 min at room temperature, unspecific binding sites blocked (1 × PBS, 5% BSA) for 1 h at room temperature. Antibodies were diluted in blocking buffer and incubated for 30 min at 37°C. Coverslips were mounted onto glass slides (Menzel) using the ProLong gold antifade reagent (Invitrogen). Secondary reagents were goat anti-rabbit or goat anti- mouse Alexa Fluor-488 or -568 (Molecular Probes) and Phalloidin Alexa Fluor-488 or -568 (Molecular Probes).

## Microscopy and image analysis

Imaging was done using an Olympus IX-81 microscope equipped with the Metamorph software, Sutter filter wheels, Photometrics CascadeII 512B camera, × 20 UPlanFL, × 40 UPlanFL, × 60 PlanApoNA1.45, or × 100 UPlanApoS NA1.4 objectives, Cobolt Jive (561 nm) and Calypso (491 nm) DPSS lasers and a TIRF condenser (Till Photonics) or on a Nikon A1R microscope equipped with an additional TIRF condenser, × 100 Nikon ApoTIRF1.49 objective, and Andor EMCCD camera. Immunofluorescence (IF) imaging of DKO MEFs was done using a LSM510 confocal micro- scope (Zeiss).

Quantification of the density of N-WASP spots in Figure 7E and F: The N-WASP channel of dual N-WASP and F-actin images was subjected to adaptive background correction in NIS Elements (Nikon) using 2.0 degrees. Images were then imported into Volocity (Perkin Elmer) and spots were detected using a 'Find Spots' algorithm with − 99% minimum offset and bright radius of 1 pixel. The cell outline was obtained by automatic segmentation using the actin channel and minimum standard deviation from image mean of '0' and a minimum object size of 5000 pixels$^2$. The number of spots detected per cell were tallied automatically and used to calculate the mean number of spots per μm$^2$.

Quantification of mRFP-Clc and Lpd-GFP spots in Figure 4G: Both channels of live-cell TIRF movies were subjected to adaptive back- ground compensation in NIS elements using 0.2 degrees and 0.5 degrees for mRFP-Clc and Lpd-GFP channels, respectively, before being filtered with a 3 × 3 median filter. All frames of a movie were then imported into Volocity for automatic object segmentation based on at least 0.1 standard deviations from the image mean with iterative size and shape exclusions. The total number of mRFP- Clc objects, Lpd-GFP objects, as well as the fraction of colocalized objects were tallied for each time point of each movie and used to calculate the mean of each group per cell.

Quantification of scission events in Figure 4I–J and Supplementary Figure S3A and B: TIRF images were subjected to background subtraction with 0.1 degrees using NIS Elements before being imported into Volocity and subjected to a fine 'remove noise' filter to improve the signal-to-noise ratio. Three 100 μm$^2$ ROIs were chosen for each cell in areas exhibiting dynamic activity of discrete mRFP-Clc spots. Each region was observed for 10 min with frames every 5 s. The disappearances of individual spots that were present for > 2 frames were observed for each channel. In total, between 150 and 350 disappearances per cell were analysed. Spots showing colocalization of mRFP-Clc together with Lpd-GFP or VASP-GFP before mRFP-Clc disappeared are presented as the percent of total disappearances ± s.e.m.

## Transferrin uptake assay

Puromycin selected control shRNA or Lpd knockdown HeLa cells were plated on coverslips, serum starved for 1 h in phenol red-free MEM, 5 mg/ml BSA and then incubated with 25 μg/ml Alexa488- transferrin (Invitrogen) for 10 min. Cells were washed twice with ice-cold PBS and once with acid wash buffer (Na-Acetate 0.2 M; NaCl 0.2 M; pH 5.3) for 2 min followed by two PBS washes and fixation in 4% PFA. To measure transferrin receptors on the cell surface, the cells were incubated with 25 μg/ml Alexa488-transfer- rin (Invitrogen) for 45 min on ice, washed three times with PBS and fixed with 4% PFA. Alexa488 in control and Lpd knockdown cells were imaged on a widefield microscope using the same exposure settings, the outline of the cells hand traced and the intensity per area quantified using Metamorph.

## Transmission electron microscopy

Puromycin selected control shRNA or Lpd knockdown HeLa cells were plated in 15 cm dishes, serum starved o/n, stimulated for 2 min with 2 ng/ml EGF, washed with ice-cold PBS, 0.1 M cacody- late buffer, and fixed with 2.5% glutaraldehyde in 0.1 M cacodylate buffer. Cells were detached gently using a cell scraper, pelleted and further processed for transmission electron microscopy as described in Taylor *et al* (2012). Images of 20 cells each for control and knockdown from two independent experiments were analysed by blinded observers.

### Proteomic screen

Lpd was *in vitro* translated using [35]S-methionin (Amersham), pBS-Lpd and the TNT Quick coupled transcription/translation system (Promega) and used to overlay a human fetal brain protein array (hEX; Source Biosciences/ImaGenes) according to the instruction of the manufacturer. Positive hits were detected using a STORM Phosphorimager (Molecular Dynamics).

### Far western blot and SPOTS scan array

Western blots of purified GST-Lpd fragments or custom-made SPOTS scan peptide arrays (Cancer Research UK services) were overlaid as described (Niebuhr *et al*, 1997) with purified MBP-Endo1/2/3-SH3 or MBP and MBP was detected with anti-MBP antibodies and the ECL kit (Pierce).

### Supplementary data

Supplementary data are available at *The EMBO Journal* Online (http://www.embojournal.org).

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

## Acknowledgements

We thank Ari Helenius (ETH Zuerich), Sasha Sorkin (University of Pittsburgh), Pietro De Camilli (Yale University), Frank Gertler (MIT), and Connie Cepko (Harvard) for sharing plasmids and antibodies. Pietro De Camilli (Yale University) for providing the DynDKO MEFs. Frank Gertler (MIT) for generous support with the protein array screen. Pat Caswell and Jim Norman (Beatson Institute, Glasgow) for protocols and advice on the EGFR uptake assay. DS is supported by a BHF grant (RE/08/003). This work was supported by grants from BBSRC (BB/G00319X/1; BB/F011431/1; BB/J000590/1) and Wellcome Trust (077429/Z/05/Z; 082907/Z/07/Z).

*Author contributions*: AV, DS, GV-B, CB, A-LL, UP, and MK performed experiments and analysed data. AV and MK wrote the manuscript. All authors discussed the results and contributed to the manuscript.

## Conflict of interest

The authors declare that they have no conflict of interest.

curvature by two newly identified structure-based mechanisms. *EMBO J* **25:** 2889–2897

Matsuda T, Cepko CL (2004) Electroporation and RNA interference in the rodent retina *in vivo* and *in vitro*. *Proc Natl Acad Sci USA* **101:** 16–22

Merrifield CJ, Qualmann B, Kessels MM, Almers W (2004) Neural Wiskott Aldrich Syndrome Protein (N-WASP) and the Arp2/3 complex are recruited to sites of clathrin-mediated endocytosis in cultured fibroblasts. *Eur J Cell Biol* **83:** 13–18

Michael M, Vehlow A, Navarro C, Krause M (2010) c-Abl, Lamellipodin, and Ena/VASP proteins cooperate in dorsal ruffling of fibroblasts and axonal morphogenesis. *Curr Biol* **20:** 783–791

Milosevic I, Giovedi S, Lou X, Raimondi A, Collesi C, Shen H, Paradise S, O'Toole E, Ferguson S, Cremona O, De Camilli P (2011) Recruitment of endophilin to clathrin-coated pit necks is required for efficient vesicle uncoating after fission. *Neuron* **72:** 587–601

Mim C, Cui H, Gawronski-Salerno JA, Frost A, Lyman E, Voth GA, Unger VM (2012) Structural basis of membrane bending by the N-BAR protein endophilin. *Cell* **149:** 137–145

Niebuhr K, Ebel F, Frank R, Reinhard M, Domann E, Carl UD, Walter U, Gertler FB, Wehland J, Chakraborty T (1997) A novel proline-rich motif present in ActA of Listeria monocytogenes and cytoskeletal proteins is the ligand for the EVH1 domain, a protein module present in the Ena/VASP family. *EMBO J* **16:** 5433–5444

Orth JD, Krueger EW, Weller SG, McNiven MA (2006) A novel endocytic mechanism of epidermal growth factor receptor sequestration and internalization. *Cancer Res* **66:** 3603–3610

Perera RM, Zoncu R, Lucast L, De Camilli P, Toomre D (2006) Two synaptojanin 1 isoforms are recruited to clathrin-coated pits at different stages. *Proc Natl Acad Sci USA* **103:** 19332–19337

Philippar U, Roussos ET, Oser M, Yamaguchi H, Kim HD, Giampieri S, Wang Y, Goswami S, Wyckoff JB, Lauffenburger DA, Sahai E, Condeelis JS, Gertler FB (2008) A Mena invasion isoform potentiates EGF-induced carcinoma cell invasion and metastasis. *Dev Cell* **15:** 813–828

Posor Y, Eichhorn-Gruenig M, Puchkov D, Schoneberg J, Ullrich A, Lampe A, Muller R, Zarbakhsh S, Gulluni F, Hirsch E, Krauss M, Schultz C, Schmoranzer J, Noe F, Haucke V (2013) Spatiotemporal control of endocytosis by phosphatidylinositol-3,4-bisphosphate. *Nature* **499:** 233–237

Pula G, Krause M (2008) Role of Ena/VASP proteins in homeostasis and disease. *Handb Exp Pharmacol* **186:** 39–65

Ringstad N, Nemoto Y, De Camilli P (1997) The SH3p4/Sh3p8/SH3p13 protein family: binding partners for synaptojanin and dynamin via a Grb2-like Src homology 3 domain. *Proc Natl Acad Sci USA* **94:** 8569–8574

Rubinson DA, Dillon CP, Kwiatkowski AV, Sievers C, Yang L, Kopinja J, Rooney DL, Ihrig MM, McManus MT, Gertler FB, Scott ML, Van Parijs L (2003) A lentivirus-based system to functionally silence genes in primary mammalian cells, stem cells and transgenic mice by RNA interference. *Nat Genet* **33:** 401–406

Schuske KR, Richmond JE, Matthies DS, Davis WS, Runz S, Rube DA, van der Bliek AM, Jorgensen EM (2003) Endophilin is required for synaptic vesicle endocytosis by localizing synaptojanin. *Neuron* **40:** 749–762

Sigismund S, Argenzio E, Tosoni D, Cavallaro E, Polo S, Di Fiore PP (2008) Clathrin-mediated internalization is essential for sustained EGFR signaling but dispensable for degradation. *Dev Cell* **15:** 209–219

Sigismund S, Woelk T, Puri C, Maspero E, Tacchetti C, Transidico P, Di Fiore PP, Polo S (2005) Clathrin-independent endocytosis of ubiquitinated cargos. *Proc Natl Acad Sci USA* **102:** 2760–2765

Soubeyran P, Kowanetz K, Szymkiewicz I, Langdon WY, Dikic I (2002) Cbl-CIN85-endophilin complex mediates ligand-induced downregulation of EGF receptors. *Nature* **416:** 183–187

Suetsugu S, Gautreau A (2012) Synergistic BAR-NPF interactions in actin-driven membrane remodeling. *Trends Cell Biol* **22:** 141–150

Tagawa A, Mezzacasa A, Hayer A, Longatti A, Pelkmans L, Helenius A (2005) Assembly and trafficking of caveolar domains in the cell: caveolae as stable, cargo-triggered, vesicular transporters. *J Cell Biol* **170:** 769–779

Taylor MJ, Lampe M, Merrifield CJ (2012) A feedback loop between dynamin and actin recruitment during clathrin-mediated endocytosis. *PLoS Biol* **10:** e1001302

Taylor MJ, Perrais D, Merrifield CJ (2011) A high precision survey of the molecular dynamics of mammalian clathrin-mediated endocytosis. *PLoS Biol* **9:** e1000604

Tsujita K, Suetsugu S, Sasaki N, Furutani M, Oikawa T, Takenawa T (2006) Coordination between the actin cytoskeleton and membrane deformation by a novel membrane tubulation domain of PCH proteins is involved in endocytosis. *J Cell Biol* **172:** 269–279

Verstreken P, Koh TW, Schulze KL, Zhai RG, Hiesinger PR, Zhou Y, Mehta SQ, Cao Y, Roos J, Bellen HJ (2003) Synaptojanin is recruited by endophilin to promote synaptic vesicle uncoating. *Neuron* **40:** 733–748

Vieira AV, Lamaze C, Schmid SL (1996) Control of EGF receptor signaling by clathrin-mediated endocytosis. *Science* **274:** 2086–2089

Warren RA, Green FA, Enns CA (1997) Saturation of the endocytic pathway for the transferrin receptor does not affect the endocytosis of the epidermal growth factor receptor. *J Biol Chem* **272:** 2116–2121

Wu M, Huang B, Graham M, Raimondi A, Heuser JE, Zhuang X, De Camilli P (2010) Coupling between clathrin-dependent endocytic budding and F-BAR-dependent tubulation in a cell-free system. *Nat Cell Biol* **12:** 902–908

Yamazaki T, Zaal K, Hailey D, Presley J, Lippincott-Schwartz J, Samelson LE (2002) Role of Grb2 in EGF-stimulated EGFR internalization. *J Cell Sci* **115:** 1791–1802

Yarar D, Waterman-Storer CM, Schmid SL (2005) A dynamic actin cytoskeleton functions at multiple stages of clathrin-mediated endocytosis. *Mol Biol Cell* **16:** 964–975

