## [Review Process File · The EMBO Journal]

Manuscript EMBO-2012-83383

Endophilin, Lamellipodin, and Mena Cooperate to Regulate F-actin-dependent EGF-receptor Endocytosis

Anne Vehlow, Daniel Soong, Gema Vizcay-Barrena, Cristian Bodo, Ah-Lai Law, Upamali Perera and Matthias Krause

Corresponding author: Matthias Krause, Kings College London

Review timeline:

Submission date:	24 September 2012
Editorial Decision:	05 November 2012
Revision received:	08 March 2013
Editorial Decision:	03 April 2013
Revision received:	28 August 2013
Accepted:	30 August 2013

Transaction Report:

Editor: Anke Sparmann

1st Editorial Decision

05 November 2012

Thank you for submitting your research manuscript (EMBOJ-2012-83383) to our editorial office. It has now been seen by three referees and their comments are provided below.

The reviewers appreciate your study and are in general supportive of publication in The EMBO Journal. However, they all agree that the functional importance of Lpd in EGFR endocytosis and CCP scission needs to be more firmly established. This is especially crucial since the effects observed after Lpd over-expression and knock-down are relatively modest. Furthermore, specificity for EGFR is not established. Overall, the referees raise a number of important technical and conceptual issues that have to be addressed by a significant amount of additional experimentation. Since this appears feasible based on their constructive suggestions, I would like to invite you to submit an extended and suitably revised manuscript to The EMBO Journal that attends to the expressed criticism in full. I should add that it is our policy to allow only a single major round of revision and that it is therefore important to address the raised concerns at this stage. Please do not hesitate to contact me should any particular argument require further clarification.

When preparing your letter of response to the referees' comments, please bear in mind that this will form part of the Review Process File, and will therefore be available online to the community. For more details on our Transparent Editorial Process, please visit our website: <http://www.nature.com/emboj/about/process.html>

We generally allow three months as standard revision time. As a matter of policy, competing manuscripts published during this period will not negatively impact on our assessment of the conceptual advance presented by your study. However, we request that you contact the editor as soon as possible upon publication of any related work, to discuss how to proceed. Should you

foresee a problem in meeting this three-month deadline, please let us know in advance and we may be able to grant an extension.

Thank you for the opportunity to consider your work for publication. I look forward to your revision.

REFEREE REPORTS

Referee #1

The authors have previously shown that Lamellipodin (Lpd) regulates actin dynamics via Ena/VASP proteins during lamellipodia formation. In this manuscript, they have investigated its implication in endocytosis. They now show that Lpd interacts with the EGF receptor and regulates its endocytosis. They also show that lpd binds to endophylin, a BAR domain containing protein involved in endocytic vesicle scission. The authors conclude that Lpd regulates actin polymerization via Ena/VASP downstream of endophylin, thus driving endocytic vesicle scission.

The first part of the manuscript illustrating the interaction of Lpd with the EGF receptor and the SH3 domain of endophylin is clear and convincing. The second part illustrating the functional importance of Lpd in EGF receptor endocytosis is much less convincing. That actin polymerization regulates endocytosis is not new. A new aspect would be to convincingly show that actin polymerization regulates vesicle fission. However, this is not fully demonstrated.

Specific comments.

1. To illustrate the functional importance of Lpd in EGFR endocytosis, the authors have used a classical cell surface biotinylation assay performed on cell depleted in Lpd or overexpressing this protein. However, the effects are mild (a 20% increase or decrease in EGF up-take). It might be more convincing to show kinetics of EGF endocytosis, at least in the first experiments, rather than measuring a 5 min time point of internalization. Because Lpd interacts with EGFR receptor (Figure 4), it would be important to show that the endocytosis of other receptors (such as transferrin receptors or LDL receptors) is not affected. It could also be important to illustrate better that Lpd is associated with clathrin-coated pits enriched in EGF-receptor (Figure 4A). This latter aspect would be critical for an accurate quantification of scission events (Figure 7G, H).
2. The authors claim that Lpd and thus actin polymerization is involved in vesicle scission. This is primarily based on the work showing an interaction between Lpd and endophylin (Figure 1-3) and the localization of Lpd-GFP to clathrin-coated pits detaching from the membrane (Figure 7G, H). Time-lapse video microscopy would be needed to convincingly show that Lpd is involved in scission. What are the kinetics of formation of coated vesicles containing Lpd and devoid of Lpd? Whether actin polymerization is by itself involved in scission events is not clear.
3. The authors state at the end of the abstract that Lpd regulates actin polymerization via Ena/VASP downstream of endophylin, thus driving endocytic vesicle scission. However, there is no real evidence for this. Figure 7 A-F just shows the localization of these components (N-WASP, Ena, Lpd) with some clathrin-coated pits. To show their functional importance, it would be necessary to base such conclusions on more solid data based on time-lapse videomicroscopy.

Referee #2

In the present manuscript a role of the VASP family interactor lamellipodin (LPD) in EGFR internalization is explored. LPD, but not its close homologue RIAM1 is shown to associate to the endocytic protein endophilin through an SH3 mediated interaction. LPD and Endophilins are shown to colocalize at CCP by TIRF microscopy. It is further shown that LPD binds and colocalizes with EGFR at CCP in an EGF-independent manner. Surface biotinylation-based internalization assays are then used to show that LPD is implicated in regulating the amount of intracellular EGFR through an actin-dependent process that possibly involves VASP family members. The authors conclude that LPD acts downstream of endophilins to regulate EGFR endocytosis in a VASP-dependent manner. This manuscript is well organized, nicely and neatly presented. The majority of the experiments are overall well executed.

There are however a number of technical and conceptual issues that need to be address for the story to be compelling.

An important set of conclusion about the role of LPD in EGFR endocytosis is based on an assays that is far from providing unequivocal results. The authors used surface labeling to subsequently measured, after 20 min of EGF stimulation, the amount of intracellular EGFR. This assay, however is not suitable to follow the very early step of Clathrin mediated internalization. CCP and CCV formation occurs within few second from stimulation. After 20 min fast and slow recycling processes significantly contribute to determine the amount of intracellular EGFR preventing the authors to reach any unequivocal conclusions on early internalization steps. Other assays using 125 I-labelled ligand or TIRF based assays would be better suited to explore the early step of CME. This is relevant as a key contention of this work is the essential role of LPD in the early step of EGFR CME. Also even by sticking to surface biotinylation assays, more careful time course would be needed.

They authors employs also TIRF microscopy to image LPD localization. However this approach is not exploited to assess whether removal of LPD alter the extent and dynamic of EGFR into CCP or to determinen the requirement of endophilin for LPD CCP localization.

Other relevant issues that should be addressed are:

LPD is claimed to act through VASP family members based on the fact that a VASP binding defective mutant does not increase the amount of intracellular EGFR. However, to support this contention, which is central to the whole work, it should be shown whether VASP family members localize to CCP and CCV, whether interference with VASP expression (either using shRNA or the FPPP mito construct that the authors have successfully used in the past) alters EGFR early internalization steps.

LPD is shown to associate with EGFR but it is unclear whether this interaction is essential for mediating its endocytic function. Is LPD affecting other CME processes, first and foremost transferrin receptor internalization? TfR internalization at variance with EGFR endocytosis is a constitutive process and exploring this aspect may provide specific functional insight into LPD roles in endocytosis.

Endophilin binds and recruits synaptojanin at late steps of CCV formation to promote clathrin uncoating. Is LPD interfering with this interaction? or with clathrin uncoating?

Additional minor point

Figure 1A-please show inputs lanes

Figure 1B please show the amounts of immunoprecipitated endophilin 3

Figure 1C is LPD localized to tubules induced by isolated BAR domain or by other BAR or F-bar containing proteins?

Figure 4A-B please quantify the extent of cellular colocalization

Figure 4F EGF stimulation appears to reduced the amount of LPD associated with EGFR. is the effect reproducible? what is the reason for this?

Figure 5-6 what is the effect on EGFR intracellular amount if both endophilin and LPD are simulatenously knocked down? Similarly, the authors speculate reasonably that N-WASP/ARP2/3 axis and LPD/VASP axis may work in concert to mediate actin dependent internalization. What is the effect of removal of both N-Wasp and VASP family members on EGFR and TfR internalization?

Figure 7F It is unclear whether the increase in N-WASP spots coincide with an increased of CCP/CCV ?

Referee #3

The authors analyze the role of lamellipodin (Lpd) in the internalization of the EGFR via clathrin-coated pits and indicate that this protein cooperates with endophilin and ENA/Vasp in inducing endocytic vesicle scission.

The data convincingly show that Lpd binds endoA3 and that these proteins colocalize. Also, the

domains involved in the interaction between these two proteins are carefully defined. Using similar approaches, the authors also show that Lpd colocalizes and forms a complex with the EGFR, and suggest that Lpd regulates EGFR endocytosis via F-actin polymerization.

This hypothesis is based on the effects of the drug LatB (which inhibits actin polymerization) and on Lpd depletion experiments. However, under these conditions, the endocytosis of the EGFR is inhibited by only 20-25%. While statistically significant, this is a small effect that should be commented upon by the authors and, more important, should be discussed in light of the mechanism that is presumably involved in the remaining 80% of the EGFR endocytic process.

The proposal that Lpd is required for CCPs scission is central in this paper and is indeed an interesting aspect of this report. However, to my judgment, it is important to show the phenotype of Lpd KD cells to further support this notion. One would expect that in parallel to the decrease in EGFR internalization (Figure 5G e 5H) one should observe (and show here) a defect in scission, which would be presumably reflected by an accumulation in CCPs connected to the plasma membrane.

Moreover, to support the role of Lpd in EGFR internalization and the role of the Lpd interaction with ENA/Vasp, it would be important also to show that in Lpd KD cells the overexpression of the Lpd wt can rescue the phenotype, whereas the overexpression of the Lpd-F/A does not (confirming the crucial role of the binding to Ena/Vasp).

Finally, as the authors indicate that Lpd recruits ENA/Vasp to the leading edge of the cells regulating lamellipodia protrusion (Krause 2004), they should show this Lpd-dependent recruitment of ENA/Vasp also at the CCPs.

Minor points

- Define the abbreviations CME, CCPs
- page 11, one line before the last, (Figure 5G and 5I) should read (Figure 7G and 7I).

1st Revision - authors' response

08 March 2013

Referee #1

The authors have previously shown that Lamellipodin (Lpd) regulates actin dynamics via Ena/VASP proteins during lamellopodia formation. In this manuscript, they have investigated its implication in endocytosis. They now show that Lpd interacts with the EGF receptor and regulates its endocytosis. They also show that lpd binds to endophylin, a BAR domain containing protein involved in endocytic vesicle scission. The authors conclude that Lpd regulates actin polymerization via Ena/VASP downstream of endophylin, thus driving endocytic vesicle scission.

The first part of the manuscript illustrating the interaction of Lpd with the EGF receptor and the SH3 domain of endophylin is clear and convincing. The second part illustrating the functional importance of Lpd in EGF receptor endocytosis is much less convincing.

>See our answer to specific comment one of this referee 1.

That actin polymerization regulates endocytosis is not new.

>It has been reported in several publications that actin polymerization may play a role in clathrin-mediated endocytosis in mammalian cells but its role is controversial (see: Anitei & Hoflack, 2012; Boucrot et al, 2006; Boulant et al, 2011; Ferguson et al, 2009; Fujimoto et al, 2000; Galletta & Cooper, 2009; Lamaze et al, 1997; Taylor et al, 2011; Wu et al, 2010; Yarar et al, 2005). However, whether actin polymerization has a role in EGFR uptake, which is inducible (in contrast to clathrin-mediated constitutive receptor endocytosis) has not been addressed.

A new aspect would be to convincingly show that actin polymerization regulates vesicle fission. However, this is not fully demonstrated.

> It has been shown by Itoh et al Dev Cell 2005 that actin polymerization supports endocytic vesicle

scission. Furthermore, this is supported by recent experiments by Taylor, Lampe, and Merrifield PLOS Biol 2012, which show that acute ablation of F-actin polymerization by LatB led to a 50% decrease in the incidence of scission in TIRF imaging based experiments. We found that knockdown of Lpd in cells devoid of dynamin increases the number of arrested clathrin-coated pits per area suggesting that Lpd functions to support vesicle scission. Our new electron microscopy experiments showing that Lpd knockdown increases the number of invaginated, omega shaped and tubulated CCPs in HeLa cells stimulated with 2 ng/ml EGF for 2 min (see new Figure 7G) further supports our view that Lpd has a role in vesicle scission.

Specific comments.

1. To illustrate the functional importance of Lpd in EGFR receptor endocytosis, the authors have used a classical cell surface biotinylation assay performed on cell depleted in Lpd or overexpressing this protein. However, the effects are mild (a 20% increase or decrease in EGF up-take).

> This referee might have missed that the effect of overexpression on EGFR uptake in the biochemistry uptake assay is indeed not modest since EGFR uptake is increased by 51% when Lpd is overexpressed and cells are stimulated with 2 ng/ml EGF (see figure 5D).

In addition, we have now done additional experiments to further explore the role of Lpd during early time points of EGFR endocytosis using the biotin biochemistry uptake assay. We found that Lpd knockdown decreases EGFR uptake at 2 minutes by approximately 56% and at 5 minutes by 29% after stimulation with 2 ng/ml EGF (see new Figures 5F and S3C). A reduction by 29-56% is not modest since it should have a significant effect on overall EGF receptor uptake.

It might be more convincing to show kinetics of EGF endocytosis, at least in the first experiments, rather than measuring a 5 min time point of internalization.

> We have now tested the role of Lpd at 2 minutes and 5 minutes of EGFR uptake (see above).

Because Lpd interacts with EGFR receptor (Figure 4), it would be important to show that the endocytosis of other receptors (such as transferrin receptors or LDL receptors) is not affected.

> We have now performed additional experiments and measured transferrin uptake in control and Lpd knockdown HeLa cells. We found that Lpd knockdown does not reduce uptake of transferrin (see new Figure S4A and S4B) suggesting that Lpd has a specific role in induced uptake of the EGF receptor and does not function in constitutive clathrin-mediated receptor endocytosis.

In addition, as mentioned above we have used TIRF imaging of control and Lpd knockdown cells to explore the role of Lpd in clathrin-mediated and clathrin-independent EGFR endocytosis. The results that clathrin-mediated endocytosis is reduced and non-clathrin-mediated uptake of the EGFR is increased (see new Figure S4C and S4D) suggests that Lpd may link the EGF receptor to CCPs and has a specific role for EGFR endocytosis similar to what has been reported for the specific role of Grb2 and CALM in EGFR uptake (Jiang et al., MBOC 2003; Huang et al., JBC 2004).

It could also be important to illustrate better that Lpd is associated with clathrin-coated pits enriched in EGF-receptor (Figure 4A).

> The co-localization of mCherry-Lpd and EGFR-GFP shown in Figure 4A is substantial considering that we expect that Lpd is only recruited to CCPs just before scission as shown in our new Figure 4G and 4H. The quantification of colocalization is done manually on many movies with hundreds of scission events. Due to time limitations for this revision we have decided to analyse a colocalization of VASP to CCPs just before scission (see new Figure 4I and 4J) and of EGFR with CCPs in the presence or absence of Lpd (see new Figure S4C and S4D).

This latter aspect would be critical for an accurate quantification of scission events (Figure 7G, H).

> To more firmly establish the role of Lpd in CCP scission we have used transmission electron microscopy to quantify the number of shallow, invaginated, omega shaped, and tubulated CCPs in Lpd knockdown and control HeLa cells that were starved and stimulated with 2 ng/ml EGF for 2 minutes. We observed that in the Lpd knockdown cells more invaginated, omega shaped, and tubulated CCP's accumulated providing further evidence that Lpd contributes to vesicle scission

(new Figure 7G).

2. The authors claim that Lpd and thus actin polymerization is involved in vesicle scission. This is primarily based on the work showing an interaction between Lpd and endophylin (Figure 1-3) and the localization of Lpd-GFP to clathrin-coated pits detaching from the membrane (Figure 7G, H). Time-lapse video microscopy would be needed to convincingly show that Lpd is involved in scission. What are the kinetics of formation of coated vesicles containing Lpd and devoid of Lpd? Whether actin polymerization is by itself involved in scission events is not clear.

> It has been shown by Itoh et al Dev Cell 2005 that actin polymerization supports endocytic vesicle scission. Furthermore, this is supported by recent experiments by Taylor, Lampe, and Merrifield PLOS Biol 2012 that show that acute ablation of F-actin polymerization by LatB led to a 50% decrease in the incidence of scission.

> Furthermore, as suggested by this referee we have used TIRF imaging of control and Lpd knockdown cells to explore the role of Lpd in clathrin-mediated and clathrin-independent EGFR endocytosis. We imaged control and Lpd knockdown HeLa cells also expressing EGFR-GFP and mRFP-clathrin light chain that were stimulated with 2 ng/ml EGF. We observed that uptake of EGFR by clathrin mediated endocytosis is reduced by 36% suggesting that Lpd indeed plays an important role in EGFR uptake. In addition, Lpd's role in EGFR endocytosis might be bigger since the reduction in EGFR uptake by clathrin-mediated endocytosis is accompanied by a compensatory increase in non-clathrin-mediated EGFR uptake by 25% (see new Figure S4C and S4D).

3. The authors state at the end of the abstract that Lpd regulates actin polymerization via Ena/VASP downstream of endophylin, thus driving endocytic vesicle scission. However, there is no real evidence for this. Figure 7 A-F just shows the localization of these components (N-WASP, Ena, Lpd) with some clathrin-coated pits. To show their functional importance, it would be necessary to base such conclusions on more solid data based on time-lapse videomicroscopy.

> We agree with the referee that time-lapse video-microscopy is the appropriate experiment to shed more light on the role of Lpd and Ena/VASP in CCP uptake. We performed live-cell imaging of HeLa cells expressing mRFP-clathrin light chain and GFP-VASP.

We observed that VASP colocalized with clathrin just before scission in 77% of CCP uptake events suggesting that VASP is indeed recruited to CCPs at the right time to support CCP uptake. We attempted to test the role of Ena/VASP in EGFR uptake using the biotin assay: We used Ena/VASP deficient knockout MEFs (MV-D7 cells, Bear et al Cell 2002) since there are three Ena/VASP protein family members and it is technically very challenging to knockdown all three proteins simultaneously. We found an anti-EGFR antibody that works on mouse EGFR for the sandwich ELISA (the assay with the human HeLa cells uses a human EGFR specific antibody for the ELISA). However, the MV-D7 Ena/VASP MEFs express only low levels of EGFR and even after scaling up the experiment to one 15 cm tissue culture plate per time point the amount of EGFR was still too low to quantitatively measure the dependence on Ena/VASP proteins for the uptake of EGFR.

Nevertheless, overexpression of the Lpd mutant that cannot bind Ena/VASP proteins (LpdF/A) failed to increase EGFR uptake in the biotin assay (in contrast to wild-type Lpd) indicating that Lpd functions via Ena/VASP proteins to regulate EGFR endocytosis. Taken together, we can only conclude that Ena/VASP mediates the effect of Lpd in EGFR uptake but cannot provide additional evidence for a direct role of Ena/VASP proteins in clathrin-mediated endocytosis. Therefore, we have amended the abstract to a more careful statement to reflect this.

Referee #2

In the present manuscript a role of the VASP family interactor lamellipodin (LPD) in EGFR internalization is explored. LPD, but not its close homologue RIAM1 is shown to associate to the endocytic protein endophilin through an SH3 mediated interaction. LPD and Endophilins are shown to colocalize at CCP by TIRF microscopy. It is further shown that LPD binds and colocalizes with EGFR at CCP in an EGF-independent manner. Surface biotinylation-based internalization assays are than used to show that LPD is implicated in regulating the amount of intracellular EGFR through an actin-dependent process that possibly involves VASP family members.

The authors conclude that LPD acts downstream of endophilins to regulate EGFR endocytosis in a VASP-dependent manner.

This manuscript is well organized, nicely and neatly presented. The majority of the experiments are overall well executed.

There are however a number of technical and conceptual issues that need to be address for the story to be compelling.

An important set of conclusion about the role of LPD in EGFR endocytosis is based on an assays that is far from providing unequivocal results. The authors used surface labeling to subsequently measured, after 20 min of EGF stimulation, the amount of intracellular EGFR. This assay, however is not suitable to follow the very early step of Clathrin mediated internalization. CCP and CCV formation occurs within few second from stimulation. After 20 min fast and slow recycling processes significantly contribute to determine the amount of intracellular EGFR preventing the authors to reach any unequivocal conclusions on early internalization steps. Other assays using 125 I-labelled ligand or TIRF based assays would be better suited to explore the early step of CME.

This is relevant as a key contention of this work is the essential role of LPD in the early step of EGFR CME. Also even by sticking to surface biotinylation assays, more careful time course would be needed.

> We have now done additional experiments to explore the requirement of Lpd during early time points of EGFR endocytosis using the biotin biochemistry uptake assay. We found that Lpd knockdown decreases EGFR uptake at 2 minutes by approximately 56% and at 5 minutes by 29% after stimulation with 2 ng/ml EGF (see new Figures 5F and S3C) suggesting that Lpd indeed has an important role in the early steps of EGFR CME.

> To more firmly establish the role of Lpd in CCP scission, an early step of EGFR CME, we have used transmission electron microscopy to quantify the number of shallow, invaginated, omega shaped, and tubulated CCP's in Lpd knockdown and control HeLa cells that were starved and stimulated with 2 ng/ml EGF for 2 minutes. We observed that in the Lpd knockdown cells more invaginated, omega shaped, and tubulated CCP's accumulated providing further evidence that Lpd contributes to early steps of EGFR CME (new Figure 7G).

They authors employs also TIRF microscopy to image LPD localization. However this approach is not exploited to assess whether removal of LPD alter the extent and dynamic of EGFR into CCP or to deterimen the requirement of endophilin for LPD CCP localization.

>As suggested by this referee we now have used TIRF imaging of control and Lpd knockdown cells to explore the role of Lpd in clathrin-mediated and clathrin-independent EGFR endocytosis. We imaged control and Lpd knockdown HeLa cells also expressing EGFR-GFP and mRFP-clathrin light chain that were stimulated with 2 ng/ml EGF. We observed that uptake of EGFR by clathrin-mediated endocytosis is reduced by 36% suggesting that Lpd indeed plays an important role in EGFR uptake. In addition, Lpd's role in EGFR endocytosis might be bigger since the reduction in EGFR uptake by clathrin-mediated endocytosis is accompanied by a compensatory increase in non-clathrin-mediated EGFR uptake by 25% (see new Figure S4C and S4D).

Other relevant issues that should be addressed are:

LPD is claimed to act through VASP family members based on the fact that a VASP binding defective mutant does not increase the amount of intracellular EGFR. However, to support this contention, which is central to the whole work, it should be shown whether VASP family members localize to CCP and CCV, whether interference with VASP expression (either using shRNA or the FPPP mito construct that the authors have successfully used in the past) alters EGFR early internalization steps.

> We agree with the referee that time-lapse video-microscopy is the appropriate experiment to shed more light on the role of VASP in CCP uptake. We now performed live-cell imaging of HeLa cells expressing mRFP-clathrin light chain and GFP-VASP.

We observed that VASP colocalized with clathrin just before scission in 77% of CCP uptake events suggesting that VASP is indeed recruited to CCPs at the right time to support CCP uptake. We attempted to test the role of Ena/VASP in EGFR uptake using the biotin assay: We used

Ena/VASP deficient knockout MEFs (MV-D7 cells, Bear et al Cell 2002) since there are three Ena/VASP protein family members and it is technically very challenging to knockdown all three proteins simultaneously. However, the MV-D7 Ena/VASP MEFs express only low levels of EGFR and even after scaling up the experiment to one 15 cm tissue culture plate per time point the amount of EGFR was still too low to quantitatively measure the dependence on Ena/VASP proteins for the uptake of EGFR.

Nevertheless, overexpression of the Lpd mutant that cannot bind Ena/VASP proteins (LpdF/A) failed to increase EGFR uptake in the biotin assay (in contrast to wild-type Lpd) indicating that Lpd functions via Ena/VASP proteins to regulate EGFR endocytosis. Taken together, we can only conclude that Ena/VASP mediates the effect of Lpd in EGFR uptake but cannot provide additional evidence for a direct role of Ena/VASP proteins in clathrin-mediated endocytosis. Therefore, we have amended the abstract to a more careful statement to reflect this.

LPD is shown to associate with EGFR but it is unclear whether this interaction is essential for mediating its endocytic function. Is LPD affecting other CME processes first and foremost transferrin receptor internalization? TjR internalization at variance with EGFR endocytosis is a constitutive process and exploring this aspect may provide specific functional insight into LPD roles in endocytosis.

> We have now performed additional experiments and measured transferrin uptake in control and Lpd knockdown HeLa cells. We found that Lpd knockdown does not reduce uptake of transferrin (see new Figure S4A and S4B) suggesting that Lpd has a specific role in induced uptake of the EGF receptor and does not function in constitutive clathrin-mediated receptor endocytosis.

In addition, we have used TIRF imaging of control and Lpd knockdown cells to explore the role of Lpd in clathrin-mediated and clathrin-independent EGFR endocytosis. The results that clathrin-mediated endocytosis is reduced and non-clathrin-mediated uptake of the EGFR is increased (see new Figure S4C and S4D) suggests that Lpd may link the EGF receptor to CCP's and has a specific role for EGFR endocytosis similar to what has been reported for the specific role of Grb2 and CALM in EGFR uptake (Jiang et al., MBOC 2003; Huang et al., JBC 2004).

Endophilin binds and recruits synaptojanin at late steps of CCV formation to promote clathrin uncoating. Is LPD interfering with this interaction? or with clathrin uncoating?

> This is an interesting question but beyond the scope of this manuscript.

Additional minor point

Figure 1A-please show inputs lanes

> We have now included the input lane for Figure 1A.

Figure 1B please show the amounts of immunoprecipitated endophilin 3

> Only 1.5% of the input lysate was loaded in the total lysate lane. We are estimating that 10 times more Lpd came down in the coIP compared to the input lane and therefore that 15% of Lpd in the cell is in complex with endophilin. However, we would expect that only a small percentage of Lpd and endophilin would be in complex at any given time since we have shown that Lpd is only recruited to CCPs just before scission were it colocalizes with endophilin.

Figure 1C is LPD localized to tubules induced by isolated BAR domain or by other BAR or F-bar containing proteins?

> Due to time constraints for this revision, we decided to focus on the TIRF experiments for VASP-GFP or EGFR-GFP with mRFP-clathrin.

Figure 4A-B please quantify the extent of cellular colocalization

> We have quantified the extent of cellular colocalization between Lpd and clathrin light chain or VASP and clathrin light chain (Figure 4G-J and S3A).

Figure 4F EGF stimulation appears to reduced the amount of LPD associated with EGFR. is the effect reproducible? what is the reason for this?

> The old Figure 4F shows Lpd-EGFR coIP after 5 minutes stimulation with 100 ng/ml EGF instead of 2 ng/ml EGF as stated in the figure legend. We would like to apologize for the wrong labeling. In the old figure the amount of Lpd that coIP's with the EGFR appears reduced. However, there is also less EGFR present in the IP due to degradation of EGFR after 100 ng/ml EGF stimulation. We have now included blots of Lpd-EGFR and EGFR-Lpd coIPs from lysates of HeLa cells stimulated with 2 ng/ml EGF for 5 minutes (as stated in the figure legend) showing that Lpd and EGFR associate both in starvation conditions and after stimulation with 2 ng/ml EGF for 5 minutes.

Figure 5-6 what is the effect on EGFR intracellular amount if both endophilin and LPD are simulatenously knocked down?

> This experiment suggested by this referee to knockdown both Lpd and endophilin 1/2/3 would be technically most likely impossible since 4 proteins (Lpd and the three endophilin family members) would have to be knocked down at the same time.

Similarly, the authors speculate reasonably that N-WASP/ARP2/3 axis and LPD/VASP axis may work in concert to mediate actin dependent internalization. What is the effect of removal of both N-Wasp and VASP family members on EGFR and TfR internalization?

> This is indeed an interesting experiment and we have used the Arp2/3 inhibitor CK666 with and without Lpd knockdown in preliminary experiments. We found that the combination of Lpd knockdown and Arp2/3 inhibition further reduced EGFR uptake suggesting that the N-WASP-Arp2/3 axis and Lpd-VASP axis work in concert and not as a linear pathway to regulate the actin cytoskeleton during EGFR CME. To substantiate these preliminary findings obtained with an inhibitor requires many additional experiments and is beyond the scope of this manuscript.

Figure 7F It is unclear whether the increase in N-WASP spots coincide with an increased of CCP/CCV ?

> To more firmly establish the role of Lpd in CCP scission we have used transmission electron microscopy to quantify the number of shallow, invaginated, omega shaped, and tubulated CCPs in Lpd knockdown and control HeLa cells that were starved and stimulated with 2 ng/ml EGF for 2 minutes. We observed that in the Lpd knockdown cells more invaginated, omega shaped, and tubulated CCPs accumulated providing further evidence that Lpd contributes to vesicle scission (now Figure 7G).

Referee #3

The authors analyze the role of lamellipodin (Lpd) in the internalization of the EGFR via clathrin-coated pits and indicate that this protein cooperates with endophilin and ENA/Vasp in inducing endocytic vesicle scission.

The data convincingly show that Lpd binds endoA3 and that these proteins colocalize. Also, the domains involved in the interaction between these two proteins are carefully defined. Using similar approaches, the authors also show that Lpd colocalizes and forms a complex with the EGFR, and suggest that Lpd regulates EGFR endocytosis via F-actin polymerization. This hypothesis is based on the effects of the drug LatB (which inhibits actin polymerization) and on Lpd depletion experiments. However, under these conditions, the endocytosis of the EGFR is inhibited by only 20-25%. While statistically significant, this is a small effect that should be commented upon by the authors and, more important, should be discussed in light of the mechanism that is presumably involved in the remaining 80% of the EGFR endocytic process.

> The referees might have missed that the effect of overexpression on EGFR uptake in the biochemistry biotin uptake assay is not modest since EGFR uptake is increased by 51% when Lpd is overexpressed and cells are stimulated with 2 ng/ml EGF (see Figure 5D).

To more firmly establish the functional importance of Lpd for EGFR endocytosis especially at early time points after EGF stimulation, we have performed additional biotin

biochemistry EGFR uptake assays. We found that Lpd knockdown decreases EGFR uptake at 2 minutes by approximately 56% and at 5 minutes by 29% after stimulation with 2 ng/ml EGF (see new Figures 5F and S3C) suggesting that Lpd plays an important role in EGFR CME.

The proposal that Lpd is required for CCPs scission is central in this paper and is indeed an interesting aspect of this report. However, to my judgment, it is important to show the phenotype of Lpd KD cells to further support this notion. One would expect that in parallel to the decrease in EGFR internalization (Figure 5G e 5H) one should observe (and show here) a defect in scission, which would be presumably reflected by an accumulation in CCPs connected to the plasma membrane.

> To more firmly establish the role of Lpd in CCP scission we have used transmission electron microscopy to quantify the number of shallow, invaginated, omega shaped, and tubulated CCP's in Lpd knockdown and control HeLa cells that were starved and stimulated with 2 ng/ml EGF for 2 minutes. We observed that in the Lpd knockdown cells more invaginated, omega shaped, and tubulated CCP's accumulated providing further evidence that Lpd contributes to vesicle scission (now Figure 7G).

Moreover, to support the role of Lpd in EGFR internalization and the role of the Lpd interaction with ENA/Vasp, it would be important also to show that in Lpd KD cells the overexpression of the Lpd wt can rescue the phenotype, whereas the overexpression of the Lpd-F/A does not (confirming the crucial role of the binding to Ena/Vasp).

> We attempted to test the role of Ena/VASP in EGFR uptake using the biotin assay: We used Ena/VASP deficient knockout MEFs (MV-D7 cells, Bear et al Cell 2002) since there are three Ena/VASP protein family members and it is technically very challenging to knockdown all three proteins simultaneously. However, the MV-D7 Ena/VASP MEFs express only low levels of EGFR and even after scaling up the experiment to one 15 cm tissue culture plate per time point the amount of EGFR was still too low to quantitatively measure the dependence on Ena/VASP proteins for the uptake of EGFR.

Finally, as the authors indicate that Lpd recruits ENA/Vasp to the leading edge of the cells regulating lamellipodia protrusion (Krause 2004), they should show this Lpd-dependent recruitment of ENA/Vasp also at the CCPs.

> We now performed live-cell imaging of HeLa cells expressing mRFP-clathrin light chain and GFP-VASP. We observed that VASP colocalized with clathrin just before scission in 77% of CCP uptake events suggesting that VASP is indeed recruited to CCPs at the same time as Lpd (see new Figure 4G-J).

Minor points

- Define the abbreviations CME, CCPs

- page 11, one line before the last, (Figure 5G and 5I) should read (Figure 7G and 7I).

> We apologize for this mistake and have changed it in the manuscript.

2nd Editorial Decision

03 April 2013

Thank you for submitting your revised manuscript for our consideration. My apologies for the slight delay in my response due to an overdue report and the intervening Easter holidays.

Your study has now been seen once more by two of the original referees, whose comments are provided below. The reviewers acknowledge that their major concerns have been addressed, and they both are in principle supportive of publication in The EMBO Journal. Nevertheless, they suggest a few minor changes that should be implemented. Please also note that referee #2 is not entirely satisfied with your response to her/his concern regarding the involvement of VASP in EGFR endocytosis. S/he requests additional experiments to corroborate this point, which seem feasible and should be included.

Please also add a statement specifying the authors' contribution.

Please remember to provide individual figure files in your final resubmission. Incidentally, the resolution of Figure 4G appears very low.

I will now return your manuscript to you for one additional round of minor revision. After that we should be able to proceed with formal acceptance and production of the manuscript!

REFEREE REPORTS

Referee #2

The manuscript has significantly improved and most of the findings have been strengthened. There are few relatively minor issues

1) It is stated that LPD knocked down reduces CME, but increases NCE (non clathrin mediated endocytosis -Fig. S4C-D). The increase in NCE of figure S4D is probably not significant and thus there is no evidence that indeed NCE is augmented following loss of Lpd. This sentence should be rephrased or more experiments should be done to support this contention (e.g there are inhibitors that can be used to impair NCE).

2) Lpd and VASP. The involvement of VASP in EGFR endocytosis remains the weakest part of the manuscript. It was asked in the first round of reviewing to support the requirement of VASP in Lpd-mediated endocytosis by showing that VASP colocalizes with clathrin (which is now shown in fig 4I). However no experiments to support the functional involvement of VASP in EGFR endocytosis has been performed. This reviewer agrees that using MVD7 cells is probably not doable due to low levels of EGFR. However, the authors of this manuscript have used extensively the mito-FPPP delocalization trick to interfere with VASP function at the plasma membrane and it would seem easy to perform this type of experiments to more directly support VASP implication on EGFR endocytosis. Alternative as suggested by another reviewer, Lpd F/A (no longer able to bind to VASP) should be used to reconstitute Lpd knockdown cells. It does remain possible that VASP is involved in the process but more experiments would be needed to corroborate this point.

Referee #3

The manuscript entitled "Endophilin and Lamellipodin Cooperate to Regulate F-Actin-Dependent Endocytosis of the EGF Receptor" by Vehlow et al. has now been extensively revised, new data and figures have been added, and all the criticisms of the referees have been satisfactorily addressed.

The manuscript is therefore much improved and it is now suitable for publication in EMBO J.

I have only a few minor comments that should however be addressed in the final version:

--Figure 4E shows WB of Lpd from lysates, immunoprecipitates using IgGs or Lpd specific antibody. All of the samples are probed with an Lpd antibody; however, the lanes refer to the specific protein run at different MWs. Since they are analysed in the same gel, the difference in MW is not obvious. The authors should explain this apparent discrepancy.

--Figure 7A-D: the labels of the panels do not correspond to the description in the legend. In particular: legend (A) says N-WASP and F-actin, and does not correspond to the Lpd label indicated in the panel; legend (B) says adaptin and F-actin, while the panel says Mena. Please amend as necessary.

--Figure 7E: It appears to me that the spots shown in panel si-Lpd-2 are larger than the spots in panel si-Lpd1. This might be due to clusters of spots, or to some defect in the fission. If the figure represents the usual phenotype under these conditions, it deserves some comment in the text.

Referee #2

The manuscript has significantly improved and most of the findings have been strengthened. There are few relatively minor issues

1) It is stated that Lpd knocked down reduces CME, but increases NCE (non clathrin mediated endocytosis -Fig. S4C-D). The increase in NCE of figure S4D is probably not significant and thus there is no evidence that indeed NCE is augmented following loss of Lpd. This sentence should be rephrased or more experiments should be done to support this contention (e.g there are inhibitors that can be used to impair NCE).

> We have removed these results from the manuscript and the figures.

2) Lpd and VASP. The involvement of VASP in EGFR endocytosis remains the weakest part of the manuscript. It was asked in the first round of reviewing to support the requirement of VASP in Lpd-mediated endocytosis by showing that VASP colocalizes with clathrin (which is now shown in fig 4I). However no experiments to support the functional involvement of VASP in EGFR endocytosis has been performed. This reviewer agrees that using MVD7 cells is probably not doable due to low levels of EGFR. However, the authors of this manuscript have used extensively the mito-FPPP delocalization trick to interfere with VASP function at the plasma membrane and it would seem easy to perform this type of experiments to more directly support VASP implication on EGFR endocytosis. Alternative as suggested by another reviewer, Lpd F/A (no longer able to bind to VASP) should be use to reconstitute Lpd knockdown cells. It does remain possible that VASP is involved in the process but more experiments would be needed to corroborate this point.

> We have generated and validated shRNAs specific against Mena and VASP and show that specifically Mena but not VASP is required for endocytosis of the EGFR at physiological concentrations of EGF (2 ng/ml). In support of this we now also show that Mena localizes to CCPs.

Referee #3

The manuscript entitled "Endophilin and Lamellipodin Cooperate to Regulate F-Actin-Dependent Endocytosis of the EGF Receptor" by Vehlow et al. has now been extensively revised, new data and figures have been added, and all the criticisms of the referees have been satisfactorily addressed.

The manuscript is therefore much improved and it is now suitable for publication in EMBO J.

I have only a few minor comments that should however be addressed in the final version:

--Figure 4E shows WB of Lpd from lysates, immunoprecipitates using IgGs or Lpd specific antibody. All of the samples are probed with an Lpd antibody; however, the lanes refer to the specific protein run at different MWs. Since they are analysed in the same gel, the difference in MW is not obvious. The authors should explain this apparent discrepancy.

> Lpd can be phosphorylated by c-Abl (Michael et al., 2010), which may cause a shift in apparent mobility. We plan to follow up a potential regulation of Lpd-EGFR interaction by c-Abl.

--Figure 7A-D: the labels of the panels do not correspond to the description in the legend. In particular: legend (A) says N-WASP and F-actin, and does not correspond to the Lpd label indicated in the panel; legend (B) says adaptin and F-actin, while the panel says Mena. Please amend as necessary.

> We would like to thank the referee for bringing this mistake to our attention and have amended the figure accordingly.

--Figure 7E: It appears to me that the spots shown in panel si-Lpd-2 are larger than the spots in panel si-Lpd1. This might be due to clusters of spots, or to some defect in the fission. If the figure represents the usual phenotype under these conditions, it deserves some comment in the text.

> This is an intriguing suggestion that Lpd knockdown may increase CCP cluster size but from the current data we can't state that the spots are significantly larger.